# Shotgun ion mobility mass spectrometry sequencing of heparan sulfate saccharides

Rebecca L. Miller [1,2,3 ✉], Scott E. Guimond[2,4], Ralf Schwörer [5], Olga V. Zubkova [5], Peter C. Tyler [5],
Yongmei Xu[6], Jian Liu[6], Pradeep Chopra [7], Geert-Jan Boons [7,8], Márkó Grabarics [9,10],
Christian Manz [9,10], Johanna Hofmann[9,10], Niclas G. Karlsson [11], Jeremy E. Turnbull[1,2,13],
Weston B. Struwe[12,13] & Kevin Pagel [9,10,13]

Despite evident regulatory roles of heparan sulfate (HS) saccharides in numerous biological processes, definitive information on the bioactive sequences of these polymers is lacking, with only a handful of natural structures sequenced to date. Here, we develop a "Shotgun" Ion Mobility Mass Spectrometry Sequencing (SIMMS[2]) method in which intact HS saccharides are dissociated in an ion mobility mass spectrometer and collision cross section values of fragments measured. Matching of data for intact and fragment ions against known values for 36 fully defined HS saccharide structures (from di- to decasaccharides) permits unambiguous sequence determination of validated standards and unknown natural saccharides, notably including variants with 3O-sulfate groups. SIMMS[2] analysis of two fibroblast growth factor-inhibiting hexasaccharides identified from a HS oligosaccharide library screen demonstrates that the approach allows elucidation of structure-activity relationships. SIMMS[2] thus overcomes the bottleneck for decoding the informational content of functional HS motifs which is crucial for their future biomedical exploitation.

[1] Copenhagen Center for Glycomics, Department of Cellular & Molecular Medicine, University of Copenhagen, Copenhagen N 2200, Denmark. [2] Centre for Glycobiology, Department of Biochemistry, Institute of Integrative Biology, University of Liverpool, Crown Street, Liverpool L69 7ZB, UK. [3] Laboratory of Cancer Biology, Department of Oncology, Medical Sciences Division, University of Oxford, Old Road Campus Research Building, Old Road Campus, Roosevelt Drive, Oxford OX3 7DQ, UK. [4] Institute for Science and Technology in Medicine, School of Medicine, Keele University, Keele, Staffordshire ST5 5BG, UK. [5] Ferrier Research Institute, Victoria University of Wellington, 69 Gracefield Road, Gracefield, Lower Hutt 5010, New Zealand. [6] Division of Chemical Biology and Medicinal Chemistry, Eshelman School of Pharmacy, University of North Carolina, Chapel Hill, NC 27599, USA. [7] Complex Carbohydrate Research Center, University of Georgia, 315 Riverbend Road, Athens, GA 30602, USA. [8] Department of Chemical Biology and Drug Discovery, Utrecht Institute for Pharmaceutical Science, and Bijvoet Center for Biomolecular Research, Utrecht University, Universiteitsweg 99, 3584 CG Utrecht, The Netherlands. [9] Freie Universitaet Berlin, Institute of Chemistry and Biochemistry, Takustrasse 3, 14195 Berlin, Germany. [10] Fritz Haber Institute of the Max Planck Society, Faradayweg 4-6, 14195 Berlin, Germany. [11] Department of Medical Biochemistry and Cell Biology, Institute of Biomedicine, Sahlgrenska Academy, University of Gothenburg, Gothenburg, Sweden. [12] Department of Chemistry, Chemistry Research Laboratory, University of Oxford, Oxford OX1 3QZ, UK. [13] These authors contributed equally: Jeremy E. Turnbull, Weston B. Struwe, Kevin Pagel. ✉email: rmiller@sund.ku.dk

Glycosaminoglycans (GAGs) are linear sulfated polysaccharides which constitute some of the most negatively charged biopolymers in nature. The heparan sulfate (HS) family of GAGs are linear polysaccharides found on the cell surface and in the extracellular matrix. GAGs are attached to core proteins in the form of proteoglycans (PGs), and exhibit great diversity in chain length and degree of sulfation patterns, creating enormous diversity for informational cues. HS chains are essential for life and orchestrate numerous biological processes including; cell migration, inflammation, anticoagulation, angiogenesis, and tumor metastasis[1]. It is generally believed that distinct structural motifs along the linear sequence of these polysaccharides serve as specific recognition sites for regulation of biological interactions. Yet only a few defined sequence motifs are known[2], perhaps the best example being the ATIII-binding pentasaccharide responsible for the anticoagulant activity of pharmaceutical heparin. This is in part due to technical limitations in the structural analysis of the large heterogenous HS chains, which is generally limited to compositional analysis of disaccharide components obtained after exhaustive enzymatic digestions[3]. Disaccharide analysis provides important "building block" analysis of sulfation and acetylation of the disaccharide units within a HS chain. However, this analysis does not provide insight into larger bioactive motifs, typically consisting of 4–10 monosaccharide units. Partial digestion and isolation of larger bioactive oligosaccharide structures (<dp10) is possible with considerable effort, but heterogeneity often prevents complete characterization[4,5]. A handful of tetrasaccharides (out of around 5000 theoretical structures) have been confidently characterized since the number of isomeric structures is limited at this size[6–9]. The heterogeneity of larger oligosaccharide structures (>dp4) impose significant challenges in complete sequencing of isomers, and even with NMR[10], chemical modification[11–13], and enzymatic deconvolution[14] <20 structures have been fully sequenced. Separation of isomeric structures is the main limitation for sequencing larger oligosaccharide fragments. Ion mobility mass spectrometry (IMMS) of glycans has demonstrated the ability to characterise isomeric arrangements of glycan building blocks (mannose, galactose, etc.), but also connectivity (glycosidic bond linkage, sialic acids) and configuration (α and β anomers)[15–18]. GAGs possess a linear sequence with no branching, and connectivity is singular to the GAG family. However, GAGs pose different challenges for IMMS analysis. GAG chain complexity is a result of heterogeneity from sulfation and epimerisation, and extended linear units. Preliminary results from our lab and others have shown that IMMS is a promising strategy for the gas phase separation of isomeric GAG oligosaccharide structures[18–24].

Here we present a method coined Shotgun Ion Mobility Mass Spectrometry Sequencing (SIMMS[2]) that relies on distinct IMMS features of smaller fragments (dp4-10) obtained by limited digestion and with reconstruction of overlapping fragments to characterize larger sequences (Fig. 1). We take advantage of a library of defined standards to build unique IMMS features that enable us to obtain unambiguous sequence information on both validated standards and unknown natural HS saccharides, and also reveal structure-activity information on FGF-inhibitory HS hexasaccharides. The SIMMS[2] sequencing method thus opens up detailed studies of bioactive motifs, aimed at deepening understanding of HS structure-function relationships and potentially permitting strategies to discover HS-based therapeutics.

## Results

**Assembling a library of HS standards.** HS is composed of a repeating uronic acid and glucosamine disaccharide unit, the uronic acid exists as either β-D-glucuronic acid (GlcA) or its epimer α-L-iduronic acid (IdoA) and can be modified with a sulfate at the $C2$ position, the glucosamine contains a N-sulfate (NS) or N-acetyl group (NAc) and can be modified with a sulfate on the $C6$ and rarely the $C3$ position. These modifications create acetylated, transition (both acetylated and sulfated) and sulfated domains (Fig. 1a). A library of standards representing features found in heparin and HS (including NS/NAc, 2O-/6O/3O-sulfation (2S/6S/3S) and IdoA/GlcA) was collected. This library encompasses a set of disaccharides isolated from heparinase-digested HS, complemented by more extended oligosaccharide structures produced by both chemoenzymatic[25–28] and chemical synthesis[29–31]. The final library included nine disaccharides, ten tetrasaccharides, six hexasaccharides, six octasaccharides, two nonasaccharides and two decasaccharides, in total providing 35 standards (#1–35), and a further standard (#36) described later (See Supplementary Table 1). Whilst containing a number of structural permutations (including NS/NAc, 2S/6S/3S, and IdoA/GlcA), this initial library was designed to be sufficiently complex to test our hypothesis that a library of defined standards can be used to develop a sequencing strategy for larger HS oligosaccharides by IMMS (Fig. 1b–d).

**IMMS separated standards display distinct characteristics.** Since we and others have shown that IMMS can resolve HS saccharide isomers as large as octasaccharides[19,20,22,23], we reasoned that it would have significant utility for HS sequencing in concert with the library of standard oligosaccharides. In IMMS molecular ions are transported through a gas-filled cell aided by a weak electric field, where they are separated according to their mass, charge, size and shape, which enables the differentiation of isomers. From the resulting arrival time distributions (ATDs), drift times can be extracted for individual components. When measured under controlled conditions, drift times can be used to calculate the collision cross section (CCS) values, a structural property related to the rotationally averaged area of a given ion. CCSs are generally given in units of ångström squared (Å[2]) and serve as molecular descriptors that can be stored in databases and used for the identification of analytes[32]. Depending on the utilized IMMS technique, CCSs can be directly calculated from the applied instrumental parameters as performed here (for details see Methods) or obtained from calibration[33]. The resulting CCS values derived from the 36 HS standards (#1–#36) and their fragments (See Supplementary Tables 1 and 2) demonstrated the ability of IMMS to distinguish between intact structures, different sized fragment ions, charge states and isomeric structures (See Supplementary Table 2).

Disaccharides—The disaccharide ΔUA-GlcNAc (#1) displayed the smallest CCS of 112Å[2], compared to disaccharide ΔUA2S-GlcNS3S6S (#29) displaying the largest CCS value of 150Å[2]. Isomeric structures ΔUA-GlcNAc6S (#2) and ΔUA2S-GlcNAc (#7) displayed different CCS values of 121 Å[2] and 124 Å[2] respectively (Fig. 2a), with ΔUA-GlcNS6S (#4) and ΔUA2S-GlcNS (#5) at CCS values of 123Å[2] and 124Å[2], respectively (See Supplementary Tables 1 and 2).

Tetrasaccharides—Among the ten defined tetrasaccharides (#9–#15, and #30–#32) in the library; four are homogenous or heterogenous tetrasaccharides containing GlcA-GlcNAc6S and IdoA-GlcNAc6S, two are heterogenous tetrasaccharides containing GlcA/IdoA-GlcNS6S, and the seventh tetrasaccharide is GlcA-GlcNS6S-IdoA2S-GlcNS6S-$R_1$. The last three tetrasaccharides contain 3O-sulfation correlating to the ATIII binding sites in porcine and bovine heparin, including ΔUA-GlcNS-IdoA2S-GlcNS3S (#30), ΔUA-GlcNS6S-GlcA-GlcNS3S6S (#31), and ΔUA-GlcNAc6S-GlcA-GlcNS3S6S (#32). GlcA and IdoA are isomeric structures that cannot be distinguished using MS/MS alone. The homogenous

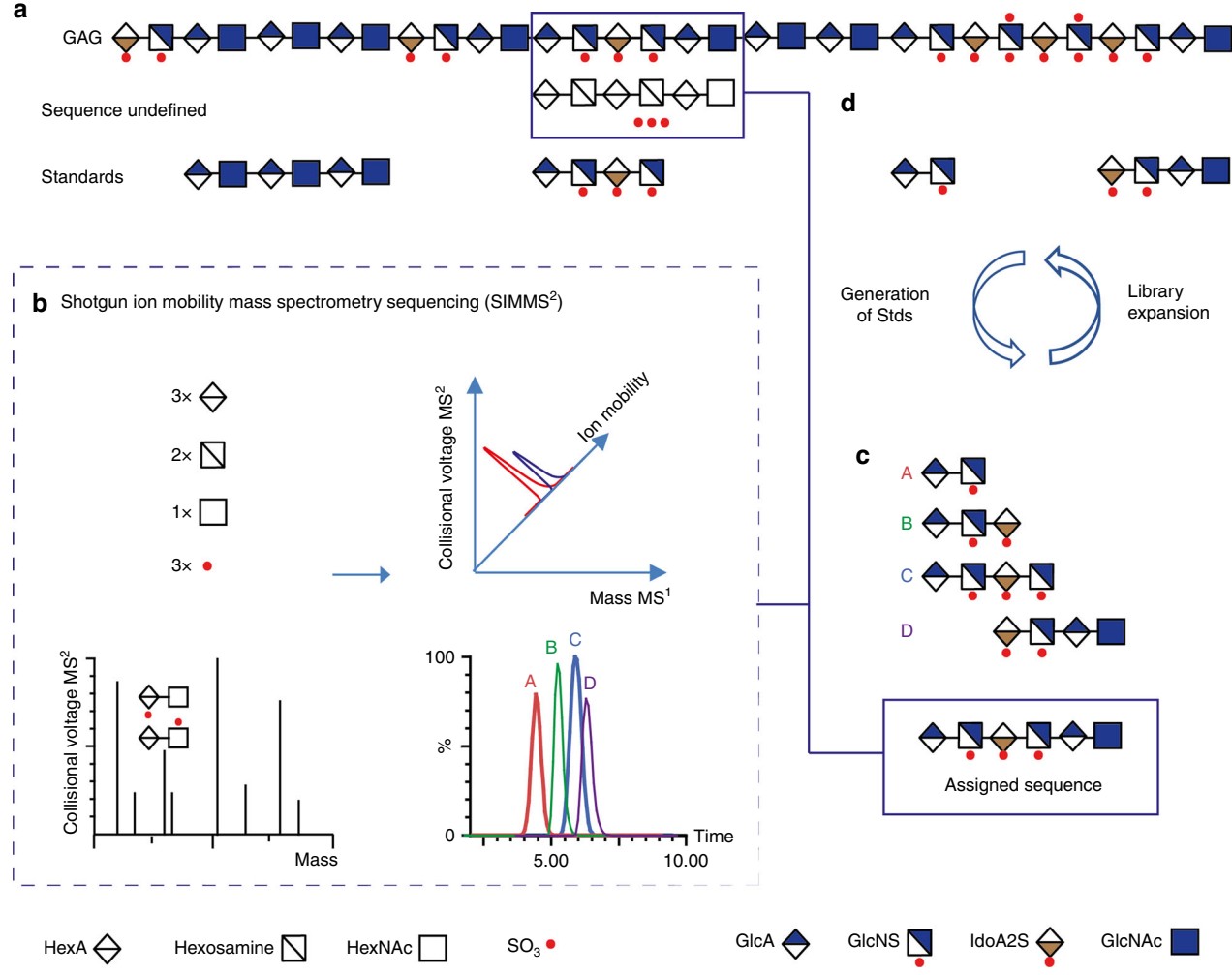

**Fig. 1 Graphic depiction of the SIMMS² strategy for de novo sequencing of HS saccharides. a** Heparin and HS contain variant domains including unmodified acetylated domains (with high levels of GlcNAc residues), transition domains (GlcNAc and GlcNS containing residues), and sulfated domains (high levels of GlcNS and *O*-sulfation). A library of shorter oligosaccharide standards were generated. **b** Illustration of MS and MS/MS to provide mass and the ability of ion mobility mass spectrometry (IMMS) to provide a third dimension that distinguishes isomers with collision cross section (CCS) values. **c** The availability of MS, MS/MS, and IMMS data from a library of known standards (**a**, **b**) then allows a strategy for unambiguous sequencing of undefined oligosaccharides based on comparison with standards. **d** Illustration of the iterative loop process for expansion of the CCS dataset necessary to further develop the SIMMS² strategy.

structures GlcA-GlcNAc6S-GlcA-GlcNAc6S-$R_1$ (**#9**) and IdoA-GlcNAc-IdoA-GlcNAc6S-$R_1$ (**#10**) can be distinguished by IMMS (Fig. 2b) with the homogenous GlcA tetrasaccharide (**#9**) displaying a CCS value of 214 Å² and IdoA tetrasaccharide (**#10**) displaying a CCS value of 242 Å² (See Supplementary Tables 1 and 2). The heterogenous mixtures GlcA-GlcNAc6S-IdoA-GlcNAc6S-$R_1$ (**#11**) and IdoA-GlcNAc6S-GlcA-GlcNAc6S-$R_1$ (**#13**) displayed CCS values of 232 Å² and 245 Å², respectively (See Supplementary Tables 1 and 2). Thus, the non-reducing end GlcA from the heterogenous tetrasaccharide structure (**#11**) displayed a more compact CCS, similar to the CCS observation in the homogenous GlcA tetrasaccharide (**#9**). The non-reducing IdoA from the heterogenous tetrasaccharide (**#13**) displayed a more extended structure closer in CCS to the homogenous IdoA tetrasaccharide (**#10**) structure than the homogenous GlcA tetrasaccharide. Crucially, all four isomeric structures (**#9**, **#10**, **#11** and **#13**) had distinct CCSs. The heterogenous GlcA-GlcNS6S-IdoA-GlcNS6S-$R_1$ (**#12**) and IdoA-GlcNS6S-GlcA-GlcNS6S-$R_1$ (**#14**) tetrasaccharides were also distinguished, with CCS values of 236 Å² and 234 Å², respectively (See

Supplementary Tables 1 and 2). As observed for the disaccharides, the *N*-sulfate group generates a more compact structure compared to saccharides with the *N*-acetyl groups, likely due to the formation of a hydrogen bond between the *N*-sulfate oxygens and the neighboring uronic acid[34,35]. The 2*O*-sulfate containing tetrasaccharide GlcA-GlcNS6S-IdoA2S-GlcNS6S-$R_1$ (**#15**) displayed a CCS value of 239 Å², which was 3 Å² larger than the same structure without the 2*O*-sulfate (**#12**), confirming that the *N*-sulfate is a dominating feature for the three-dimensional structure (See Supplementary Tables 1 and 2). The 3*O*-sulfated tetrasaccharides corresponding to the ATIII sites in porcine ΔUA-GlcNS-IdoA2S-GlcNS3S (**#30**) and ΔUA-GlcNS6S-GlcA-GlcNS3S6S (**#31**) displayed CCS values of 228 Å² and 229 Å² respectively. Whilst the tetrasaccharide corresponding to the ATIII site in bovine heparin ΔUA-GlcNAc6S-GlcA-GlcNS3S6S (**#32**) displayed a CCS value of 234 Å².

Hexasaccharides to decasaccharides—Six hexasaccharides were tested. The structure GlcA-GlcNS6S-IdoA-GlcNS6S-GlcA-GlcNS6S-$R_1$ (**#16**) is a disaccharide extension of one of the tested tetrasaccharides (**#14**) and displayed a CCS value of 259 Å².

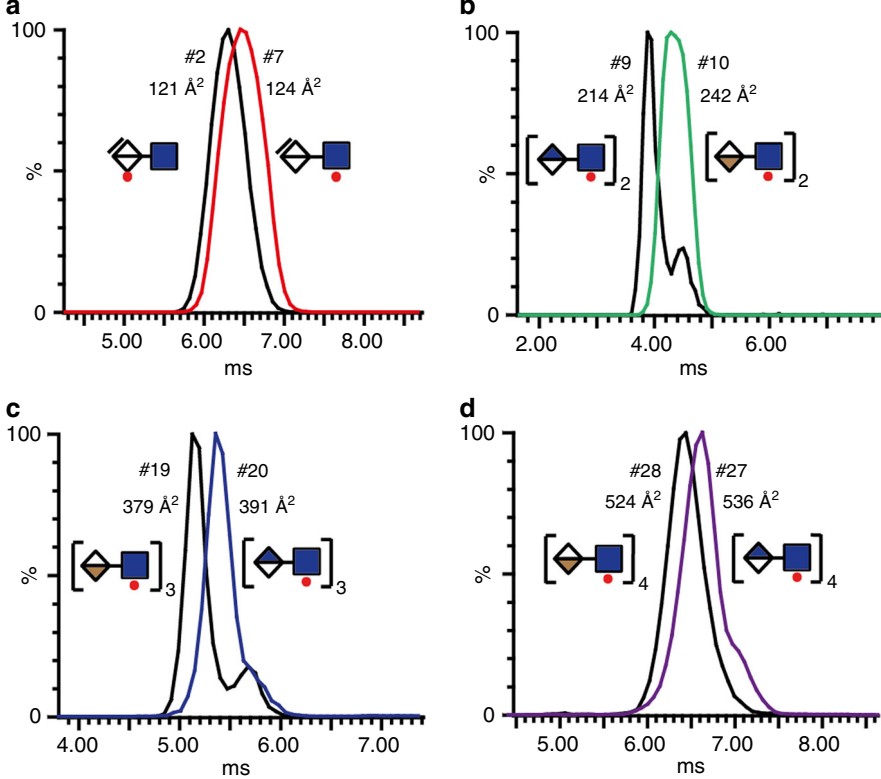

**Fig. 2 Drift tube ion mobility separation (DTIMS) of isomeric HS standards.** Intact HS standards were subjected to DTIMS to define collision cross section (CCS) values. Variant degrees of separation were observed for (**a**), Disaccharides ΔUA-GlcNAc6S (**#2**) and ΔUA2S-GlcNAc (**#7**). **b** Tetrasaccharides GlcA-GlcNAc6S-GlcA-GlcNAc6S-R$_1$ (**#9**) and IdoA-GlcNAc6S-IdoA-GlcNAc6S-R$_1$ (**#10**), where R$_1$ is (CH$_2$)$_5$NH$_2$. **c** Octasaccharides GlcNAc6S-[IdoA-GlcNAc6S]$_3$-IdoA-R$_2$ (**#19**) and GlcNAc6S-[GlcA-GlcNAc6S]$_3$-GlcA-R$_2$ (**#20**), where R$_2$ is C$_7$H$_7$O. **d** Decasaccharides GlcNAc6S-[IdoA-GlcNAc6S]$_4$-IdoA-R$_2$ (**#27**) and GlcNAc6S-[GlcA-GlcNAc6S]$_4$-GlcA-R$_2$ (**#28**), where R$_2$ is C$_7$H$_7$O. Full details of structures and CCS values are provided in Supplementary Tables 1 and 2.

The two homogenous isomeric hexasaccharide structures GlcNAc6S-[GlcA-GlcNAc6S]$_2$-GlcA-R$_2$ (**#17**) and GlcNAc6S-[IdoA-GlcNAc6S]$_2$-IdoA-R$_2$ (**#18**) displayed distinctly different CCS values of 297 Å$^2$ and 292 Å$^2$, respectively (See Supplementary Table 2). Structures G-GlcNS6S-G-GlcNS6S-I2S-GlcNS6S-R$_1$ (**#33**) and G-GlcNS6S-G-GlcNS3S-I2S-GlcNS6S-R$_1$ (**#34**) are isomeric displaying CCS values of 280 Å$^2$ and 282 Å$^2$ respectively. Structure G-GlcNS6S-G-GlcNS3S6S-I2S-GlcNS6S-R$_1$ (**#35**) displayed a CCS value of 328 Å$^2$.

Furthermore, we tested six octasaccharide structures GlcNAc6S-[UA-GlcNAc6S]$_3$-UA-R$_2$ (**#19**–**#24**) that were either homogenous with GlcA or IdoA residues, or heterogenous with two GlcA and two IdoA residues. All these isomeric octasaccharides are also distinguishable based on their distinct CCS values (Fig. 2c and See Supplementary Tables 1 and 2). Whether a GlcA or IdoA was next to the methoxyphenyl glycoside tag appeared to affect the octasaccharide structure, with the GlcA-R on the reducing end displaying a more extended structure than the IdoA-R on the reducing end.

Two decasaccharides with homogenous disaccharide structures GlcNAc6S-[IdoA-GlcNAc6S]$_4$-IdoA-R$_2$ (**#27**) and GlcNAc6S-[GlcA-GlcNAc6S]$_4$-GlcA-R$_2$ (**#28**) were also distinguished with CCS values of 536 Å$^2$ and 524 Å$^2$, respectively (Fig. 2d and See Supplementary Table 2). Finally, we also found that two nonasaccharides GlcA-GlcNS-GlcA-GlcNS-IdoA-GlcNS-GlcA-GlcNS-GlcA-R$_3$ (**#25**) and GlcA-GlcNS-GlcA-GlcNS-IdoA2S-GlcNS-GlcA-GlcNS-GlcA-R$_3$ (**#26**) differing only in 2O-sulfate on the iduronic acids can be distinguished based on their CCS values of 437 Å$^2$ and 441 Å$^2$, respectively.

**Fragmentation of defined standards**. The tetra- to decasaccharides standards were subjected to fragmentation in the trap cell of the mass spectrometer prior to separation in the IMMS cell (See Supplementary Tables 3–51). This process resulted in unique CCS values for each fragment ion where homogenous IdoA B fragment ions (as defined by Domon and Costello nomenclature[36]) resulted in more extended structures, compared to homogenous GlcA B fragment ions (See Supplementary Tables 3–51). In contrast, the hexa- to decasaccharides with a nitrophenyl tag next to the IdoA/GlcA affected the structural conformation, with IdoA Y product ions resulting in a more compact structures compared to that of GlcA. A comparison of B ions with GlcNAc6S or GlcNS6S showed that the NS group resulted in a more compact structure (See Supplementary Tables 3–18). Importantly, data obtained on internal fragments from standards demonstrates the utility of single larger standards to provide an extended set of CCS data values.

**Development of a HS sequencing method—SIMMS$^2$**. While MS and MS/MS alone do not enable unambiguous sequencing of HS chains, our IMMS studies on the library of HS standards demonstrated that IMMS could, firstly, distinguish HS structures with different lengths up to decasaccharides, secondly, determine disaccharide composition and order, and thirdly, distinguish many isomeric structures. These collective abilities provide the necessary data for complete structural identification of a large number of important structures. We therefore hypothesized that with sufficient information on fragmentation patterns and CCS values from relevant standards, it would be possible to

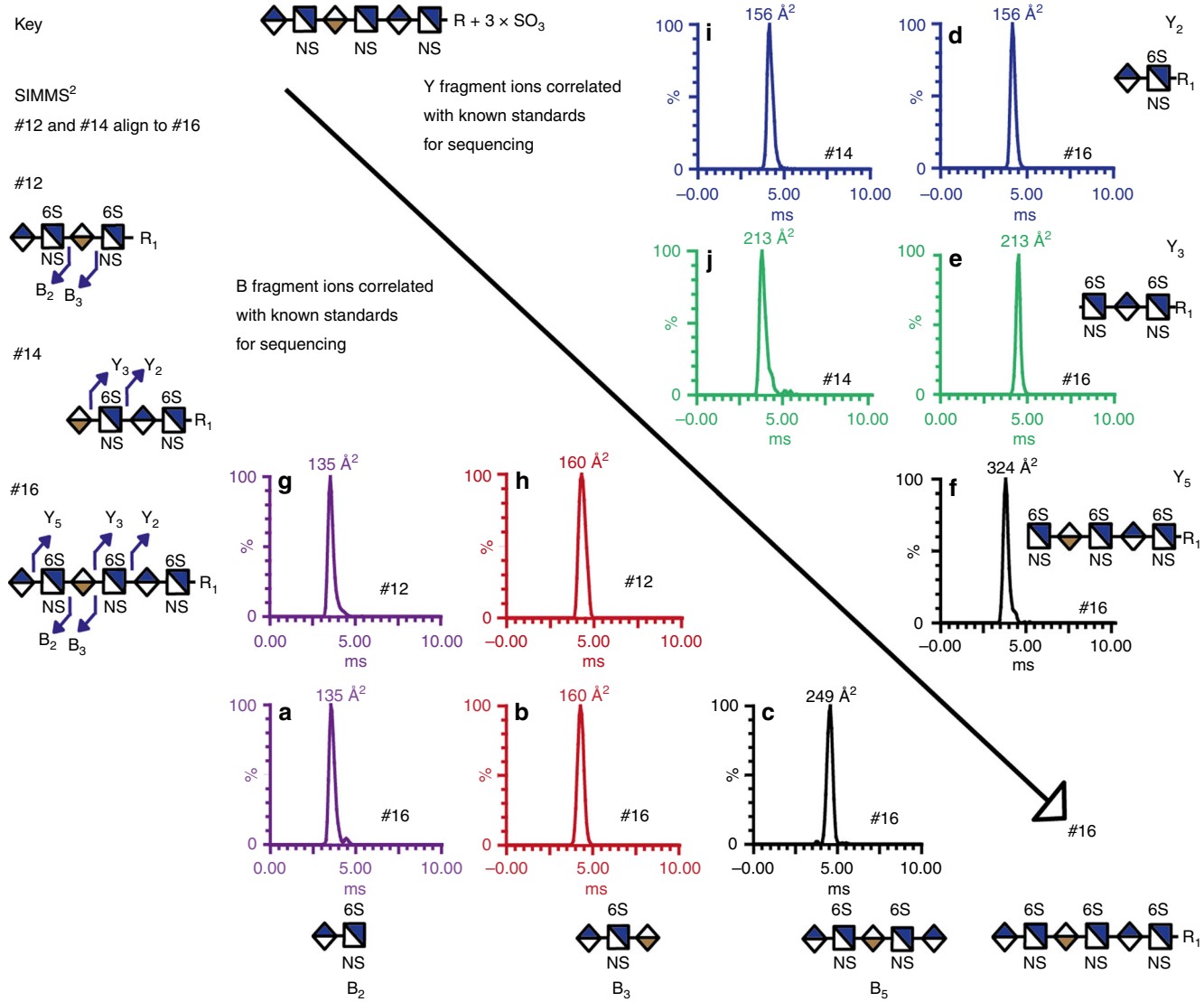

**Fig. 3 Using the SIMMS² method for sequencing a hexasaccharide.** Overlaying two tetrasaccharide standard structures (**#12** and **#14**) provides complete sequence coverage of the hexasaccharide (**#16**). Each structure was fragmented in the trap of the mass spectrometer and analyzed using drift tube ion mobility mass spectrometry (DTIMS), resulting in accurate collision cross section (CCS) values. A comparison of overlapping fragment ions displayed the same CCS value between the two tetra- and hexasaccharides, providing a three-dimensional (MS, MS/MS and IMMS) sequencing method. **a–f** The hexasaccharide displays the DTIMS data to be determined. **g, h** Tetrasaccharide **#12** displayed CCS values from B ions; $B_2$—135 Å² and $B_3$—160 Å² matched CCS values observed in the hexasaccharide (**a–c**). **i, j** Tetrasaccharide **#14** displayed CCS values from Y ions; $Y_2$ 156 Å² and $Y_3$ 213 Å² matched the CCS values observed in the hexasaccharide (**d–f**).

unambiguously sequence unknown HS structures using data from overlapping standards in a shotgun IMMS strategy—designated SIMMS².

To test this hypothesis, we used SIMMS² to sequence the hexasaccharide GlcA-GlcNS6S-IdoA-GlcNS6S-GlcA-GlcNS6S-$R_1$ (**#16**) (Fig. 3). This approach exploits data from two tetrasaccharides GlcA-GlcNS6S-IdoA-GlcNS6S-$R_1$ (**#12**) and IdoA-GlcNS6S-GlcA-GlcNS6S-$R_1$ (**#14**) that represent all the possible monosaccharides contained in this hexasaccharide. Tetrasaccharide **#12** generated CCS of $B_2$ 135 Å² and $B_3$ 160 Å² matching the CCS data $B_2$ 135 Å² and $B_3$ 160 Å² from the hexasaccharide (**#16**), confirming that the reducing ends were identical. Tetrasaccharide **#14** generated CCS of $Y_2$ 156 Å² and $Y_3$ 213 Å², matching with $Y_2$ 156 Å² and $Y_3$ 213 Å² from the hexasaccharide **#16** confirming that the non-reducing ends are identical (Fig. 3, and See Supplementary Tables 4 and 6 display the full set of B, Y, C, and

Z ions). In addition, with the aid of CCS values of fragments ions created by loss of a sulfate group from the two tetrasaccharides **#12** and **#14**, we were able to sequence the hexasaccharide structure **#16** (See Supplementary Fig. 1 and See Supplementary Tables 28, 29 and 31).

In another example the structures of a nonamer (**#25**) and the nonamer with a single 2O-sulfate (**#26**) were compared by SIMMS² (Fig. 4a and See Supplementary Tables 15–16) demonstrating that this strategy could determine the position of the single 2O-sulfate on the IdoA residue in nonasaccharide **#26**. This supports the hypothesis that SIMMS² has the potential to serve as a method for sequencing extended glycosaminoglycan structures.

**Applying SIMMS² to define ATIII and FGF-regulatory epitopes.** It is generally observed that the majority of biological

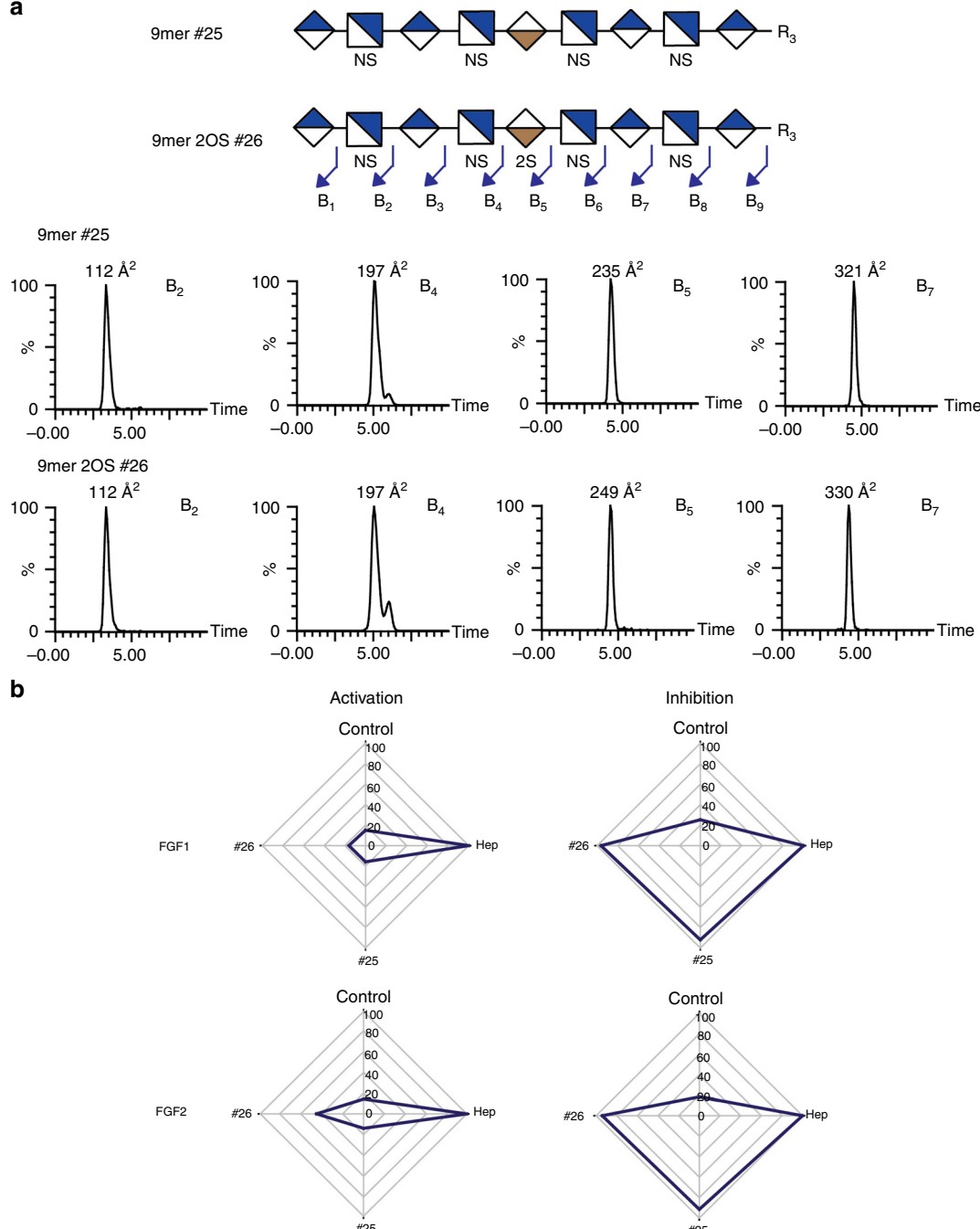

**Fig. 4 MS² sequencing of two 9mer saccharides.** HS structures **#25** and **#26** were fragmented in the trap and separated through IMMS to create a defined set of CCS values for each fragment. IMMS of fragments (**a**) represent B ion fragments from **#25**, and **#26**, indicating that the 2O-sulfate is on the IdoA at position 5. Fragment ions from B/Y/C/Z are displayed in Supplementary Tables 15 and 16. **b** Radar chart illustration of BaF3 cell activation and inhibition assays of HS structures **#25** and **#26**. Activation (cell proliferation) assays were performed with fibroblast growth factor 1 (FGF1) or FGF2 (1 ng/mL) and standards **#25/26**. Heparin (3 μg/mL) was used as positive control, while FGF1 or FGF2 alone was used as negative control. Inhibition assays were performed with the same compounds (3 μg/mL) in the presence of a sub-maximal dose of heparin (0.1 μg/mL). Cell proliferation results were expressed as a percentage of heparin activity set as 100%. Data for panel b are presented as mean ± standard deviation, $n = 3$. Source data are provided as a Source Data file.

interactions of HS chains are conferred by motifs within the range of 4–10 monosaccharide residues. The classic high affinity antithrombin III binding site responsible for the anticoagulant activity in pharmaceutical heparin is an important example;[37,38] therefore CCS values were determined for the enzymatically depolymerized natural structures (**#29–#32**) and synthesized structures (**#33–35**) that correlated with the porcine and bovine

ATIII binding sites (See Supplementary Tables 19 to 25). SIMMS² was used to sequence isomeric structures **#33** and **#34** (See Supplementary Table 1) that differed only in the presence of either a 6O-sulfate or a 3O-sulfate on a single glucosamine residue (See Supplementary Fig. 2). B fragment ions from **#33** and **#34** were distinguishable: the B₄ fragment ion of the 6O-sulfated isomer (**#33**) has a CCS value of 225 Å², whereas the B₄ fragment

of the isomer carrying 3$O$-sulfation (#34) has a CCS value of 220 Å$^2$ (See Supplementary Fig. 2 and Supplementary Tables 23 and 24). This data clearly demonstrates the ability of SIMMS$^2$ to determine structures containing 3$O$-sulfate groups.

In addition, we used FGF1 and FGF2 regulation as a relevant model to demonstrate sequencing capabilities for natural isolated HS structures. We used BaF3 cell assays[39] to evaluate activation and inhibition of FGFs by HS compounds. First, we validated the assay with heparin and the nonamer compounds #25/26, which interestingly showed that the single 2$O$-sulfate group introduced in #26 provides weak partial FGF2 activation, while neither structure showed inhibitory activity (Fig. 4b).This is noteworthy since previous results demonstrated that $NS$ and 2$OS$ are required for FGF2 binding, while 6$OS$ is required for optimal promotion of FGF2 cellular signaling[40,41]. The nonasaccharide #26 lacks 6$OS$ and yet partially promotes FGF2 signaling, demonstrating the utility of complete characterization of sequences by SIMMS$^2$ in deciphering cues for FGF bioactivities. We then obtained HS by-products from pharmaceutical heparin production[42], and used limited heparinase III digestion to produce a range of oligosaccharide size fractions, which were initially separated by size exclusion chromatography (SEC). The dp8 and dp6 fractions were further separated through strong anion exchange (SAX)-HPLC and selected fractions were tested for bioactivity in the BaF3 assay (See Supplementary Fig. 3a–b). Only the late eluting dp8 subfractions (more highly sulfated) induced FGF1 and FGF2 activation (See Supplementary Fig. 3a–b); however, we were unable to separate these fractions further into single entities due to low abundance and high heterogeneity of these fractions. In contrast, the dp6 fractions separated nicely into single oligosaccharides, and two of these (b$^1$ and b$^3$) showed substantial FGF2 inhibitory activity (See Supplementary Fig. 3b). The two inhibitory dp6 fractions were separated further by cetyltrimethylammonium-SAX (CTA-SAX)[43], and the purity assessed by MS and disaccharide analysis, confirming that each represented a single hexasaccharide structure designated #HS1 (b$^1$) and #HS2 (b$^3$) (See Supplementary Fig. 4a–c).

Employing the SIMMS$^2$ method and taking disaccharide analysis into account, the potential hexasaccharide structures were limited and sequences for #HS1 and #HS2 could be provisionally assigned as ΔUA-GlcNS-IdoA2S-GlcNS-UA-GlcNAc and ΔUA-GlcNS-IdoA2S-GlcNS6S-UA-GlcNAc6S, respectively (See Supplementary Tables 52–69). Both proposed hexasaccharides display the same non-reducing end (ΔUA-GlcNS-IdoA2S), and our IMMS data confirmed that the first three non-reducing monosaccharides indeed had the same CCS values (Fig. 5 and See Supplementary Tables 59 and 68). The current standard library we employed is necessarily limited to a subset of HS structures. To expand the library of standards to include saccharides containing a 2$O$-sulfate and a carbon-carbon double bond, we performed enzymatic digestion of the nonamer standard #26 (with one 2$O$-sulfate group), which provided the 7mer standard #36 (See Supplementary Table 1) with a single 2$O$-sulfate. The SIMMS$^2$ method was applied to the ATIII tetrasaccharide #30, septsaccharide #36 and the hexasaccharides #HS1 and #HS2 with a set of MS, MS/MS and CCS values for each ΔUA-GlcNS-IdoA2S containing structure (See Supplementary Tables 20, 26, 59, and 68). The CCS fragment ion values from standard #36 were compared to the HS purified hexasaccharide #HS1 (Fig. 5a and See Supplementary Tables 26 and 59), and the CCS data for B$_1$, B$_2$, and B$_3$, B$_4$ were the same in both structures (71, 110, 162 and 201Å$^2$, respectively; Fig. 5a and See Supplementary Tables 26 and 59). This resulted in unambiguous sequencing of hexasaccharide #HS1 and assignment of the sequence as ΔUA-GlcNS-IdoA2S-GlcNS-GlcA-GlcNAc. A comparison of #36, and #HS1 with #HS2 revealed that the B$_4$ exhibited different CCS values demonstrating the position of the predicted 6$O$-sulfate

groups and thus assigning the hexasaccharide structure of #HS2 as ΔUA-GlcNS-IdoA2S-GlcNS6S-GlcA-GlcNAc6S (Fig. 5a, See Supplementary Fig. 4c, and Tables 26, 59, and 68). A comparison of tetrasaccharide #30 (from the ATIII binding site) to #HS2 confirmed B$_1$, B$_2$, and B$_3$ fragment ions. The B$_4$ ion of #30 and #HS2 are isomers, with the former containing a 3$O$-sulfate and a CCS value of 218 Å$^2$ and the latter a 6$O$-sulfate with a CCS of 225 Å$^2$ (Fig. 5 and See Supplementary Tables 20 and 68). We retested the homogenous #HS1 and #HS2 hexasaccharides in the BaF3 cell assay and found that they inhibit FGF2-dependent cell proliferation with similar IC50 values of 638 and 566 nM, respectively (Fig. 5b). The #HS1 and #HS2 hexasaccharides differ in only 6$O$-sulfation. Previous studies have shown that 2$O$-sulfate is required for FGF2 binding, 6$O$-sulfate is additionally required for receptor activation, although the detailed structural requirements for binding versus activation is poorly understood[39,41]. However, as discussed above we found that the 9mer standard #25 was unable to activate FGF2, whereas in contrast the modified 9mer standard #26 with only a single 2$O$-sulfate group sequenced by SIMMS$^2$ displayed partial but significant activation of FGF2 (Fig. 4b). A decasaccharide containing two 6$O$-sulfate groups was previously shown to have FGF2 inhibitory activity[39], and our results demonstrate two smaller hexasaccharides with none or two 6$O$-sulfate groups that exhibit similar inhibitory activity. It is thus clear that the mere presence of 2$O$-sulfate and 6$O$-sulfate is not enough to drive regulation of FGF2 activity, and further studies are now justified to pursue the detailed features of FGF-regulatory HS motifs, and will clearly be facilitated by SIMMS$^2$ methodology.

## Discussion

The SIMMS$^2$ method presented here relies on the resolving power of IM to define IMMS features of smaller fragments (dp4-10), and the reconstruction of overlapping fragments to characterize larger sequences. It was identified that both odd and even fragments are informative for SIMMS$^2$ sequencing. This method provides a substantial step forward in enabling direct sequencing of HS saccharides. A key objective for the development of the SIMMS$^2$ strategy was to demonstrate sequencing of bioactive HS motifs, since this has been a major bottleneck for advancement of the field. We were able to demonstrate identification of hexasaccharide inhibitors with both 2$O$-sulfate and 6$O$-sulfate groups which were notably selective for FGF2 versus FGF1, suggesting the possibility of developing FGF-specific regulatory compounds. We also exploited SIMMS$^2$ to sequence an enzymically modified 9mer and showed that the addition of a single 2$OS$ group conferred partial activation of FGF2 (but not FGF1), despite lacking any 6$O$-sulfation. This illustrates the utility of SIMMS$^2$ for defining structure-activity relationships for GAG saccharides and emphasize its potential for the discovery of bioactive motifs with pharmacological potential[44–46].

SIMMS$^2$ is a methodology relying on access to accurate CCS values from standards, and while we demonstrated proof-of-concept for sequencing HS through synthetic standards, further development of SIMMS$^2$ for complete analysis of all HS structures will require access to a larger library of standards, generation of a comprehensive CCS dataset, and improved strategies to isolate and obtain homogeneous and heterogenous oligosaccharides. Separation of isomeric glycans in some cases is possible, whereas some isomeric glycans display similar CCS values[47]. In the present study, the same phenomenon was observed, and is a recognized limitation of IM/MS methodology in its current form. Thus, to ensure correct oligosaccharide sequence identification, a comparison of multiple CCS fragment values should be performed, and the interpretation should include information from MS/MS and disaccharide analysis. We anticipate further extension to other

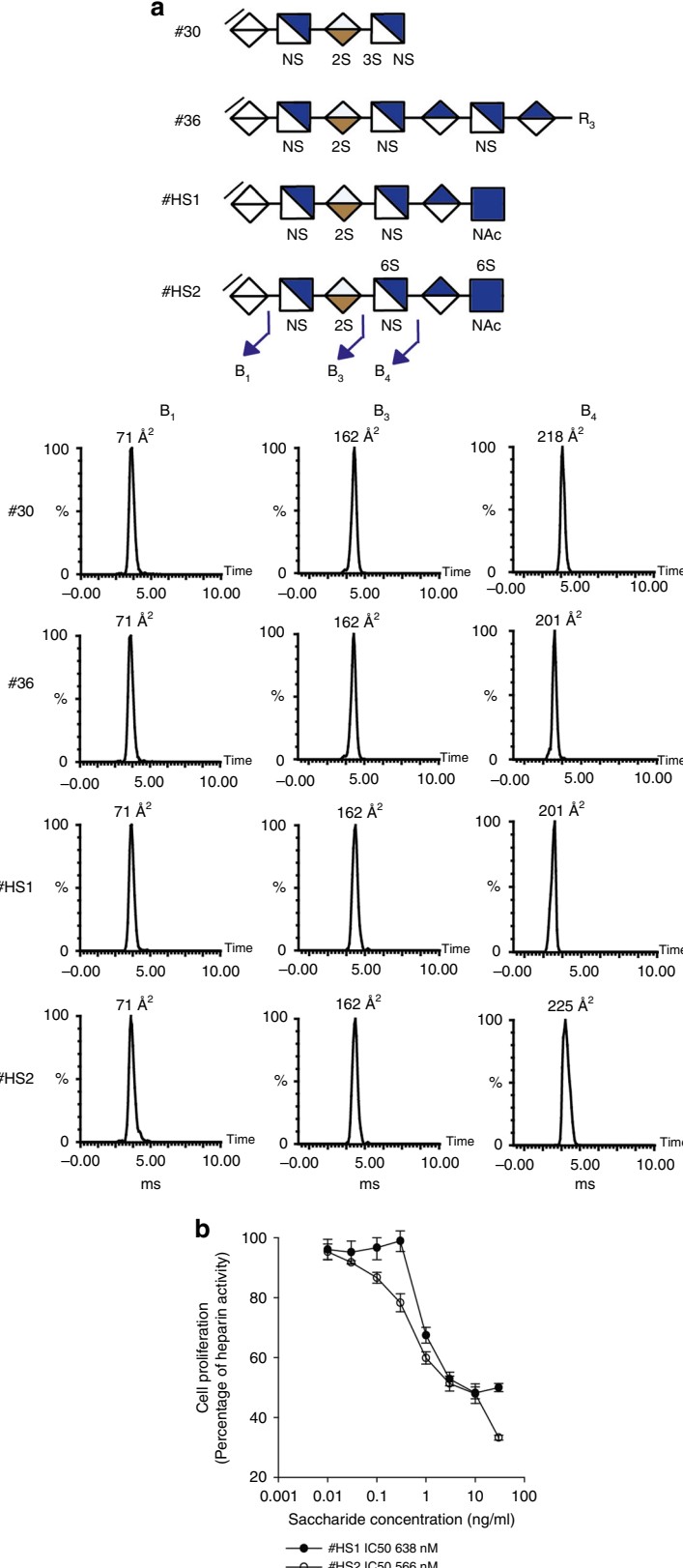

**Fig. 5 Applying the SIMMS² method for sequencing HS motifs displaying FGF bioactivity. a** SIMMS² sequencing of hexasaccharides #HS1 and #HS2 through the comparison of B ion fragment CCS values from standard **#30** and **#36**. **b** BaF3 FGF2 inhibition assay with a sub-maximal dose of heparin (0.1 μg/mL) in the presence of serially diluted #HS1 (black circle with line) and #HS2 (white circle with line) saccharides. Data for (**b**) are presented as mean ± standard deviation, $n = 3$. Source data are provided as a Source Data file.

simpler GAGs will also be possible using a similar strategy as suitable defined standards become available. Recent advances in chemical[30,31] and chemoenzymatic[25-28] synthesis of GAG saccharides are providing expanding resources of structures to support building such standards and CCS datasets. Purification of GAG oligosaccharides from tissues with current chromatography methods is notoriously challenging due to the complex mixtures obtained from such heterogenous sources. However, an emerging source for standards may be cell lines with engineered capacities for producing limited subsets of GAG features[3,48], which should simplify separation and isolation of homogeneous and heterogenous oligosaccharides. In particular, the GAGOme method employing the robust Chinese hamster ovary (CHO) cell line to engineer and produce specifically restricted repertoires of GAGs with distinct features can help reduce complexities. These CHO cells can also be used to produce distinct GAGs on secreted proteins or by metabolic priming on glycosides, which may offer unique access to defined standards required for developing the SIMMS² method.

A further aspect for expanding utility of the SIMMS² method is interpretation and analysis of spectra for HS and potentially other GAG saccharides. Proteomics software packages perform poorly with GAG spectra, since they commonly use averaging algorithms of monoisotopic peaks and charge states. However, solutions are beginning to emerge[49,50]. For example, Hogan et al.[49] uses a library of every possible theoretical fragment of a GAG saccharide for each charge state and compares the theoretical isotopic distribution with observed spectral patterns. Data sharing provides the resources for the development of improved algorithms for GAG analysis, and our dataset from 36 oligosaccharides are publicly available in UniCarbDR[51] under MIRAGE guidelines[52]. These resources will require continuous updating and expansion, and they should be allied with effective software development. Generation of sequence data on expanding numbers of HS saccharides, and links to relevant protein targets, will also provide important information to support molecular modelling[53] and interaction studies[54] which will aid in defining their mechanisms of action in biological regulation.

The principles of SIMMS² sequencing rely on the generation of accurate CCS values and the resolving power of IMMS instruments. The linear drift tube (DT)IMS employed here is the only IMMS technique that enables the direct determination of accurate CCSs without a need for a calibration procedure. Oligosaccharide CCS values in this study were determined using a single DTIMS instrument. Reproducibility, uncertainty and instrument variability has been addressed previously in an interlaboratory study[55]. For fatty acids and metabolites an average error of $0.27 \pm 0.18\%$ and $0.44 \pm 0.28\%$ was observed. Peptides and proteins showed the largest error with the stepped field method, $0.53 \pm 0.44\%$ and $0.68 \pm 0.36\%$, respectively. Our CCS data from 36 defined oligosaccharide structures can now be applied as calibrants to determine CCS values on other IMMS instruments, potentially including traveling wave (TW)IMS, trapped (T)IMS and cyclic (c) IMS. Moreover, the utility to distinguish isomers relies on the IMMS resolving power, which can be expected to continue to improve with advances in instrumentation, and this could help alleviate this current limitation of the methodology. For example, if the CCS of two isomeric ions differ by 2%, a resolving power of 64 (corresponding to a plate number of 22 500) is required to separate them by DTIMS with a peak-to-peak resolution of 0.75. SIMMS² as a concept is entirely compatible with other MS labs, as state-of-the-art commercial and custom-built DTIMS instruments can achieve the required resolving power. Furthermore, current developments in cIMS devices working with traveling waves enable capabilities of achieving much higher levels of resolving power, heralding immense potential for advances in IMMS-based de novo sequencing strategies for HS and other GAGs.

Whilst sequencing bioactive HS motifs is the most pressing need, and is now enabled by the use of the SIMMS² strategy, intact HS chains are large, heterogenous, and composed of multiple bioactive motifs. Defining more details of sequences of intact HS chains remains a future challenge in which SIMMS² may make a contribution. Only one example of intact GAG chain sequencing has been reported[56], where a short CS chain on the proteoglycan bikunin with only 4O-sulfation was sequenced. A second attempt was applied on the short DS chains on decorin[57]; significant heterogeneity was observed and isomeric structures were unresolved, but partial information on chain length and level of sulfation was acquired. Nevertheless, we envision that limited digestion of HS or GAG chains with multiple site-specific enzymes will generate unique sets of partially overlapping fragments that can be separated and then sequenced by SIMMS²; overlaps of such fragments may enable assembly in a similar manner to the shotgun sequencing approaches used in genomics and proteomics. This could ultimately lead to insights into domain structures and higher order arrangements of intact HS and other GAG chains.

In summary, SIMMS² is a shotgun method which enables direct and unambiguous sequencing of HS oligosaccharides through assembly of overlapping fragments. The method alleviates previous limitations imposed by the isomeric nature of many HS saccharides, and we envision SIMMS² being further developed and expanded to enable sequencing of all types of GAG oligosaccharides. Its application should accelerate decoding the informational codes that underpin their functional versatility and promise significant advances in their biomedical exploitation.

## Methods

**Materials and reagents.** All chemicals were of analytical or HPLC grade purity from Sigma (Gillingham, UK) or VWR (Lutterworth, UK) unless otherwise stated. Disaccharide standards 1–8 were from Iduron (Manchester, UK), HS from IntelliHep (Liverpool, UK) and heparinases from IBEX (Montreal, Canada). Chemically synthesized structures were produced as described[31]. Structure (**#36**) was produced chemoenzymatically as described[25]. HS-2-O-sulfotransferase was expressed in Origami B(DE3) cells (Novagen) and purified by chromatography as described[58].

**Chemical synthesis of standards.** For the synthesis of standards desired building blocks were prepared following reported procedures and assembled employing modular approach[31,59]. The levulinoyl (Lev) esters of the sequences were removed by treatment with hydrazine acetate, and subjected to O-sulfation using sulfur trioxide/pyridine complex in DMF, followed by Fmoc removal with triethylamine/dichloromethane (1/4, v/v). The methyl esters and acetates were saponified with $H_2O_2$ and LiOH in THF. Next, the azide groups of the compounds were reduced under Staudinger conditions using $PMe_3$/THF (1.0 M), and the resulting amines were N-sulfated, employing a sulfur trioxide/pyridine complex in MeOH in the presence of $Et_3N$ and NaOH. The target structures were obtained by hydrogenation, over Pd/C in a mixture of $tBuOH/H_2O$ (1/1, v/v) to cleave the protecting group of the linker followed by further hydrogenation over $Pd(OH)_2$/C to remove the benzyl ethers, and finally purified by SEC over a P-2 column. All final structures were characterized by NMR spectroscopy and high resolution MS.

**Preparation of 3O-sulfated di- and tetrasaccharide standards.** The 3-O-sulfated disaccharide and 3-O-sulfated tetrasaccharide standards were prepared from corresponding oligosaccharides after heparin lyases digestion as described previously[60]. Four 3-O-sulfated standards were used in this study, including ΔUA2S-GlcNS3S6S, ΔUA-GlcNS-IdoA2S-GlcNS3S, ΔUA-GlcNS6S-GlcA-GlcNS6S, and ΔUA-GlcNAc6S-GlcA-GlcNS3S6S. To prepare ΔUA2S-GlcNS3S6S disaccharide standard, a synthetic octasaccharide with a structure of GlcNAc6S-GlcA-GlcNS6S-IdoA2S-GlcNS3S6S-IdoA2S-GlcNS6S-GlcA-pNP (pNP represents p-nitrophenyl group) was incubated with a mixture of heparin lyase I, II, and III in 50 mM phosphate buffer pH 7.0 at 37 ºC overnight. The ΔUA2S-GlcNS3S6S disaccharide was purified by a Q-Sepharose column. The structure of ΔUA2S-GlcNS3S6S disaccharide standard was confirmed by electrospray ionization mass spectrometry (ESI-MS) and by NMR as described[60]. To prepare ΔUA-GlcNS-IdoA2S-GlcNS3S, ΔUA-GlcNS6S-GlcA-GlcNS3S6S, and ΔUA-GlcNAc6S-GlcA-GlcNS3S6S, three different octasaccharide substrates with the structures of GlcNAc-GlcA-GlcNS-IdoA2S-GlcNS3S-IdoA2S-GlcNS-GlcA-pNP, GlcNAc-GlcA-GlcNS6S-GlcA-

GlcNS3S6S-IdoA2S-GlcNS6S-GlcA-pNP, and GlcNAc-GlcA-GlcNAc6S-GlcA-GlcNS3S6S-IdoA2S-GlcNS6S-GlcA-pNP, respectively, were subjected to heparin lyases digestion. Each tetrasaccharide was purified by Q-Sepharose, and the structure was confirmed by ESI-MS and NMR[60].

**CCS measurements**. Drift tube (DT)IMS and a helium buffer gas (HE) was used to directly calculate $^{DT}CCS_{He}$, designated as CCS (drift velocity of ions across the cell is proportional to mobility). CCS of standards was performed on a Synapt HDMS instrument modified with a linear drift tube IM cell as described[61,62], allowing determination of CCS values without calibration. Mass Lynx V4.1 software (Waters) is used to operate the instrument and save data files. Samples were measured at 0.05 μM in water/acetonitrile (50/50 v/v). Oligosaccharides were ionized using nano-electrospray ionization (nESI) and Pd/Pt or gold-coated borosilicate capillaries fabricated in-house. Mass spectra were acquired in negative ion mode with a capillary voltage at 0.70 kV. Standard **#10** was analyzed under soft and harsh source conditions, and demonstrated that the determined CCS value was unchanged (See Supplementary Fig. 5). To obtain product ion mobility data and produce comparable CID data for each isomer, MS/MS was performed on selected ions in the trap cell of the instrument at 3, 8, and 18 V (with argon as collision gas). CCS values were determined using the stepped-field method, i.e., by plotting the arrival times (centroid of the best-fit Gaussian) as a function of reciprocal drift voltages. Correlation coefficient ($R^2$) of the linear fits were typically ≥0.9999. ATDs were recorded at eight different voltages ranging from 50 to 120 V. Each given CCS value is the average of two to four independent measurements with deviations generally around 0.5%. Drift times were extracted and CCSs calculated using software developed in-house or manually using origin. Drift times were extracted manually where peak broadening, split peaks or peak shoulders were evident. The **a**lgorithm for **p**rocessing **i**on-mobility **d**ata (Aprid) was written in-house and used for CCS calculation in an automated fashion. Details about the code can be found in the code availability section.

**Expression of 2*O*-sulfotransferase**. The maltose binding protein (MBP)-2*O*-sulfotransferase is a fusion protein was created using a pMAL-c2x vector (New England Biolabs). The MBP was truncated at the Asn-367 residues and the 2OST contained a mutation at E359A. The linker region encodes three alanine residues (A368–A370) and contained a NotI site for cloning. The catalytic domain of chicken 2OST (D69–N356) was cloned into the vector by using the NotI and BamHI sites. The MBP-2OST was expressed in Origami B(DE3) cells (Novagen). The cells were grown in LB medium and induced using isopropyl-β-D-thiogalactopyranoside (ITGP). Cells were left to grow on a shaker at 37 °C or overnight at 18 °C. Cells were pelleted and resuspended in 25 mM Tris (pH 7.5), 500 mM NaCl, and 1 mM DTT. Sonication was used to lyse the cells. The MBP-2OST expressed fusion protein was purified through an amylose resin (New England Biolabs) and eluted from the beads using maltose[58].

**Generation of a 9-mer HS standard with 2*O*-sulfation**. The 9mer standard **#25** (500 μg) (See Supplementary Table 1) was incubated with 5 mU 2*O*-sulfotransferase in 100 mM PAPS, 50 mM MES, pH 7 at 30 °C overnight. After heat inactivation (98 °C, 4 min) products were separated by SAX-HPLC on a ProPac PA1 column (4.6 mm × 250 mm, 5 μm bead size, Dionex) using a NaCl gradient (0–1.4 M NaCl in HPLC grade water). Product with addition of a single 2*O*-sulfate was observed at 30% yield; this 9mer + 2*O*-sulfate (**#26**) was digested further with heparinase II at 1 mU/mg to generate standard **#36**.

**Pure bioactive hexasaccharide structures isolated from HS**. HS (500 mg) was dissolved in 900 μL of lyase buffer (100 mM sodium acetate, 10 mM calcium acetate) and incubated with Heparinase I (0.5 mU/mg) at 37 °C, with aliquots (300 μL) taken at 2, 4, and 8 h and heat inactivated. The aliquots were pooled and separated by SEC using preparation grade Superdex 30 beads (15 mm × 170 cm, bead size 34 μm – GE Healthcare) on an Akta FPLC system with isocratic elution in 0.5 M NH₄HCO₃ (0.5 mL/min; 232 nm). Fractions containing HS of similar dp were pooled and repeatedly freeze-dried using HPLC grade water to remove NH₄HCO₃. Dp8 and dp6 SEC fractions were selected and separated by SAX-HPLC (ProPac PA1) as described above using a 0–1.4 M NaCl gradient. The two main fractions were desalted and separated by VSCTA-SAX on a cetyl-trimethylammonium derivatized C18 column (4.6 mm × 250 mm, 5 μm bead size – Sigma) using a gradient of 0–1.5 M NH₄HCO₃, as described[43]. Each fraction was dried on a speed vac (Thermo Scientific) and subjected to DTIMS and compositional analysis.

**Compositional analysis**. HS fractions digested with 0.1 mU each of heparinases I, II, and III were incubated at 30 °C for 24 h, followed by HPLC on a Propac PA1 column with a NaCl gradient (0–1 M, 60 min) monitoring at 232 nm; peaks were evaluated by comparison with authentic standards.

**BaF3 FGF signaling assays**. The activity of HS oligosaccharides with FGF1 and FGF2 was determined using BaF3 cells transfected with FGFR1c (from David

Ornitz, Washington University, St. Louis)[39,42]. BaF3 cells ($1 \times 10^4$ cells/well) were plated into 96 well plates with recombinant FGF-1 or FGF-2 (1 ng/mL; R&D Systems) and either heparin or oligosaccharides (3 μg/mL). After 72 h 5 μL/well of 5 mg/mL MTT was added and incubated for 4 h. Formazan product was solubilized with 10% SDS/0.01 N HCl and quantified at 570 nm. For inhibition assays suboptimal heparin was added at 0.1 μg/mL, with oligosaccharides tested at 3 μg/mL Data are presented as mean ± standard deviation, $n = 3–9$.

**Reporting summary**. Further information on research design is available in the Nature Research Reporting Summary linked to this article.

## Data availability
Materials are available on request to the authors. All data generated or analyzed during this study are included in this article and/or associated Supplementary Information Tables 1–69 and uploaded to UniCarbDR. The source data underlying Figs. 4b and 5d and Supplementary Fig. 3b are provided as a Source Data file.

## Code availability
The algorithm for processing ion-mobility data (Aprid) was written in-house and used for CCS calculation in an automated fashion. It is implemented using Python 2.7 and can be operated by the user through a graphic interface devised in PyQtDesigner. The main goal of the script is to determine collision cross-sections (CCSs) for specific mass-to-charge (*m/z*) ratios. For that purpose, the script follows the mathematical approach of solving the Mason-Schamp equation. Details about the code and modules can be found online via github: https://github.com/waschbaerlauch/CCS-Calculation.

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

## Acknowledgements

This work was supported by an Oxford CRUK development fund grant (CRUK0317 to R.L.M. and W.B.S.); a Senior Research Fellowship from the Medical Research Council (G117/423), Biotechnology and Biological Sciences Research Council project grants (BB/I004343/1; BB/K02128/1; BB/MO27791), a PhD studentship from MRC/Engineering and Physical Sciences Research Council (to J.E.T.); the National Institutes of Health (HL094463-09 and HL144970 to J.L.); The National Institute of General Medical Sciences (NIGMS; P41GM103390) from US National Institute of Health (G-J.B); a Center for Research Resource grant of the National Institutes of Health (2 P41 RR005351 to G-J.B.); a Human Frontier Science Program grant (RGP0062 to G-J.B. and J.E.T.); Marsden Fund New Zealand grant (to P.C.T. and J.E.T.); the Deutsche Forschungsgemeinschaft (DFG, German Research Foundation) (Project number 372486779 – SFB 1340 to K.P.); and the Danish National Research Foundation (DNRF107).

## Author contributions

R.L.M. performed all MS analyses and interpretations. S.E.G. performed the FGF bioassay experiments and enzymatic modification of the 9mer saccharide. J.H., M.G., K.P., and W.B.S. assisted with the IMMS instrument and CCS data analysis. C.M. provided Python created software to calculate CCS values. R.S., O.V.Z., P.C.T., P.C., and G.J.B generated the chemically synthesized standards. J. L. and Y.X. generated the chemoenzymatic synthesized standards. N.G.K. contributed to upload and database storage

in UniCarbDR. R.L.M., J.E.T., K.P., and W.B.S. conceived and designed the study. R.L.M. wrote the manuscript and all authors contributed to manuscript revisions and approved the final version.

## Competing interests

The authors declare no competing interests.
