## [Peer Review File · Nature Communications]

REVIEWERS' COMMENTS:

Reviewer #1 (Remarks to the Author):

The authors made the following comments to justify their use of lower-resolution DTIMS method: "The resolving power of an instrument and the error associated with the determination of CCS (accuracy) are different. Instruments with higher resolving power do not necessarily provide better accuracy and lower uncertainties. In our method a DTIMS was employed, which does not rely on calibration. Using the stepped-field method, DTIMS provides absolute CCS values with the highest achievable accuracy."

While it is true that the lower-resolution DTIMS method may provide absolute CCS values with the highest achievable accuracy FOR A SINGLE FRAGMENT, the value of higher resolving power cannot be understated when dealing with a mixture of fragments with closely spaced CCS values. I think it is important for the authors to acknowledge this limitation.

Although I still believe that ambiguities could arise due to sulfo losses, formation of internal fragments, and difficulty in differentiating reducing and nonreducing end fragments, the authors have shown that their approach has sufficient merits and utility in GAG sequencing that outweigh this limitation.

Overall, I am satisfied with this revision, and have no further objection on its publication.

Reviewer #3 (Remarks to the Author):

I think the authors have effectively addressed my previous concerns. The revised manuscript is greatly improved and is acceptable for publication in my opinion.

Nature Communications sequencing paper NCOMMS-19-29535A

Point by point response.

Query #1: We therefore invite you to revise your paper one last time to address the remaining concerns of Reviewer#1. At the same time we ask that you edit your manuscript to comply with our format requirements and to maximise the accessibility and therefore the impact of your work.

Reviewer #1 (Remarks to the Author):

The authors made the following comments to justify their use of lower-resolution DTIMS method: "The resolving power of an instrument and the error associated with the determination of CCS (accuracy) are different. Instruments with higher resolving power do not necessarily provide better accuracy and lower uncertainties. In our method a DTIMS was employed, which does not rely on calibration. Using the stepped-field method, DTIMS provides absolute CCS values with the highest achievable accuracy."

While it is true that the lower-resolution DTIMS method may provide absolute CCS values with the highest achievable accuracy FOR A SINGLE FRAGMENT, the value of higher resolving power cannot be understated when dealing with a mixture of fragments with closely spaced CCS values. I think it is important for the authors to acknowledge this limitation.

Although I still believe that ambiguities could arise due to sulfo losses, formation of internal fragments, and difficulty in differentiating reducing and nonreducing end fragments, the authors have shown that their approach has sufficient merits and utility in GAG sequencing that outweigh this limitation.

Overall, I am satisfied with this revision, and have no further objection on its publication.

Response #1: We have added 2 pieces of additional text to stress the limitation noted by the Reviewer.

Action #1:

Line 312: Separation of isomeric glycans in some cases is possible, whereas some isomeric glycans display similar CCS values⁴⁷. In the present study, the same phenomenon was observed, **and is a recognized limitation of IM/MS methodology in its current form**. Thus, to ensure correct oligosaccharide sequence identification, a comparison of multiple CCS fragment values should be performed, and the interpretation should include information from MS/MS and disaccharide analysis.

Line 352: "Moreover, the utility to distinguish isomers relies on the IMMS resolving power, which can be expected to continue to improve with advances in instrumentation, **and this could help alleviate this current limitation of the methodology**."

Query #2: * To adhere to journal style, I suggest the following revision to the title. If you would like to suggest an alternative title, please ensure that it does not exceed 15 words and does not contain punctuation.

"Shotgun Ion Mobility Mass Spectrometry Sequencing of Heparan Sulfate Saccharides"

Response #2: Changed to suggested title from the editor.

Action #2: Title now

"Shotgun Ion Mobility Mass Spectrometry Sequencing of Heparan Sulfate Saccharides"

Query #3: * I have made some edits to the abstract. Please check that you agree with them.

Response #3: We agree with the changes and have implemented them into the paper as track changes.

Action #3: added track changes to the abstract that the editor made.

Query #4: MAIN TEXT * Please use the present tense when discussing the current work in the Introduction.

Response #4: Introduction has been checked and edited to conform.

Action #4: The introduction has been modified to keep the current work in present tense; lines 56, 57, 66, 68, 81, 85, 88, 90 and 93.

Query #5: MAIN TEXT * Please shorten all subheadings in the Results section to fewer than 60 characters including spaces.

Response #5: All headings have been edited to less than 60 characters.

Action #5:

Before (line 110): Standards separated through IMMS display distinct characteristics

After: IMMS separated standards display distinct characteristics (57 characters)

Before (line 219): Using SIMMS² for defining ATIII and FGF regulatory HS epitopes

After: Applying SIMMS² to define ATIII and FGF regulatory epitopes (60 characters)

Query #6: LANGUAGE AND STYLE * Please remove language such as "new", "novel", "for the first time", "unprecedented", etc. Novelty should be clear from the context.

Response #6: Usage of "new" and "novel" identified in the text.

Action #6: Removed uses of "new" and "novel" from the paper.

Query #7: LANGUAGE AND STYLE * Please do not use italics, bold font or underlining to convey emphasis (in both the main text and the display items).

Response #7: Italics has not been used for emphasis, only to define an Oxygen (as in 2*O*-sulfation) or Carbon (C3 position), as per standard nomenclature. No other Italic or underlining was used.

Action #7: Bold text has been removed throughout.

Query #8: LANGUAGE AND STYLE * Bold font should be used for numbering chemical compounds. Chemical abbreviations or formulae should not be bolded. This applies to both the main text and the display items.

Response #8: Compounds #1 to #36 are now in bold throughout the text.

Action #8: Numbered chemical compounds #1 to #36 are in bold.

Query #9: METHODS AND DATA * Please shorten all subheadings in the Methods section to fewer than 60 characters including spaces.

Response #9: Titles in the method section have been shortened to fewer than 60 characters

Action#9: Edits

Before (line 492): Isolation of pure bioactive hexasaccharide structures from heparan sulfate

After: Pure bioactive hexasaccharide structures isolated from HS (57 Characters)

Query #10: * In the Methods section, please provide sufficient information such that the experiments could reasonably be reproduced without reference to other papers. Instead of saying "as described previously", please describe briefly all protocols, including standard protocols and previously published protocols. Please note that there are no word limits for the Methods section. **In particular, please describe how the structures were synthesized and characterized, and how HS-2-O-sulfotransferase was produced and purified.**

Response #10: Chemical synthesis of structures and expression / purification of the 2O-sulfotransferase has been described.

Action #10:

Line 425: Chemical synthesis of standards: For the synthesis of standards desired building blocks were prepared following reported procedures and assembled employing modular approach.^{31, 59} The levulinoyl (Lev) esters of the sequences were removed by treatment with hydrazine acetate, and subjected to *O*-sulfation using sulfur trioxide/pyridine complex in DMF, followed by Fmoc removal with triethylamine/dichloromethane (1/4, v/v). The methyl esters and acetates were saponified with H₂O₂ and LiOH in THF. Next, the azide groups of the compounds were reduced under Staudinger conditions using PMe₃/THF (1.0 M), and the resulting amines were *N*-sulfated, employing a sulfur trioxide/pyridine complex in MeOH in the presence of Et₃N and NaOH. The target structures were obtained by hydrogenation, over Pd/C in a mixture of tBuOH/H₂O (1/1, v/v) to cleave the protecting group of the linker followed by further hydrogenation over Pd(OH)₂/C to remove the benzyl ethers, and finally purified by size exclusion chromatography over a P-2 column. All final structures were characterised by NMR spectroscopy and high resolution MS.

Line 437: Preparation of 3O-sulfated di- and tetrasaccharide standards. The 3-*O*-sulfated disaccharide and 3-*O*-sulfated tetrasaccharide standards were prepared from corresponding oligosaccharides after heparin lyases digestion as described previously⁶⁰. Four 3-*O*-sulfated standards were used in this study, including ΔUA2S-GlcNS3S6S, ΔUA-GlcNS-IdoA2S-GlcNS3S, ΔUA-GlcNS6S-GlcA-GlcNS3S6S, and ΔUA-GlcNAc6S-GlcA-GlcNS3S6S. To prepare ΔUA2S-GlcNS3S6S disaccharide standard, a synthetic octasaccharide with a structure of GlcNAc6S-GlcA-GlcNS6S-IdoA2S-GlcNS3S6S-IdoA2S-GlcNS6S-GlcA-pNP (pNP represents *p*-nitrophenyl group) was incubated with a mixture of heparin lyase I, II and III in 50 mM phosphate buffer pH 7.0 at 37°C overnight. The ΔUA2S-GlcNS3S6S disaccharide was purified by a Q-Sepharose column. The structure of ΔUA2S-GlcNS3S6S disaccharide standard was confirmed by electrospray ionization mass spectrometry (ESI-MS) and by NMR as described⁶⁰. To prepare ΔUA-GlcNS-IdoA2S-GlcNS3S, ΔUA-GlcNS6S-GlcA-GlcNS3S6S, and ΔUA-GlcNAc6S-GlcA-GlcNS3S6S, three different octasaccharide substrates with the structures of GlcNAc-GlcA-GlcNS-IdoA2S-GlcNS3S-IdoA2S-GlcNS-GlcA-pNP, GlcNAc-GlcA-GlcNS6S-GlcA-GlcNS3S6S-IdoA2S-GlcNS6S-GlcA-pNP, and GlcNAc-GlcA-GlcNAc6S-GlcA-GlcNS3S6S-IdoA2S-GlcNS6S-GlcA-pNP, respectively, were subjected to heparin lyases digestion. Each tetrasaccharide was purified by Q-Sepharose, and the structure was confirmed by ESI-MS and NMR⁶⁰.

Line 474: Expression of 2O-sulfotransferase. The MBP-2O-sulfotransferase is a fusion protein was created using a pMAL-c2x vector (New England Biolabs). The MBP was

truncated at the Asn-367 residues and the 2OST contained a mutation at E359A. The linker region encodes 3 alanine residues (A368–A370) and contained a NotI site for cloning. The catalytic domain of chicken 2OST (D69–N356) was cloned into the vector by using the NotI and BamHI sites. The MBP-2OST was expressed in Origami B(DE3) cells (Novagen). The cells were grown in LB medium and induced using isopropyl- β -D-thiogalactopyranoside (ITGP). Cells were left to grow on a shaker at 37°C or overnight at 18°C. Cells were pelleted and resuspended in 25 mM Tris (pH 7.5), 500 mM NaCl, and 1 mM DTT. Sonication was used to lyse the cells. The MBP-2OST expressed fusion protein was purified through an amylose resin (New England Biolabs) and eluted from the beads using maltose⁵⁸.

Query #11: * Please state the origins of the Sf9 cells used in the Methods section (ATCC number, vendor, catalogue number, as applicable).

Response #11: The later method used the Sf9 cells, the cells used to express the 2OST enzyme used in the study was Origami B(DE3) cells (Novagen). The origin of the Origami B(DE3) cells are now stated in the Method section (Line 479).

Action #11: Origami B(DE3) cells (Novagen / Sigma-Aldrich (Cat: 70837)).

Query #12: * DATA SOURCES: * In the Data Availability statement, please specify which data are available via UniCarbDR and provide the corresponding accession codes. In addition, please provide specific references to relevant supplementary items instead of referring to 'Supplementary Information files' in general. We also ask that you either deposit all raw data or explain in your Data Availability Statement why these data can only be shared on request.

Response #12: Data acquired for structures #1 to #36 and HS1 and HS2 (all structures used in the study) can be found in the UniCarb DR Database <https://unicarb-dr.biomedicine.gu.se/references/388>. Unlike proteins, accession codes don't currently exist for glycosaminoglycans. The UniCarb DR is a database curated for glycans and recently glycosaminoglycan. We're trying to encourage the depository of data for sugars here. I've previously placed data in this depository for two papers <https://unicarb-db.expasy.org/references/342> and <https://unicarb-dr.biomedicine.gu.se/references/353>.

Action #12: URL's added for UniCarb DR <https://unicarb-dr.biomedicine.gu.se/references/388>. under Data availability, line 519.

Query #13: * To ensure correct hyperlinking of the accession codes in your manuscript, please add the hyperlink or DOI in square brackets directly after the code throughout (for example, "5XRN [<http://dx.doi.org/10.2210/pdb5XRN/pdb>]", "1483958 [<https://dx.doi.org/10.5517/ccdc.csd.cc1t5m6>]", "SRP109982 [<https://www.ncbi.nlm.nih.gov/sra/?term=SRP109982>]" or "NQLW00000000 [https://www.ncbi.nlm.nih.gov/assembly/GCA_002312845.1/]").

Response #13: Added one DOI link to line 412.

Action #13: Added <http://dx.doi.org/10.2210/pdb5XRN/pdb> to line and 412.

Query #14: DISPLAY ITEMS * Please check whether your manuscript or Supplementary Information contain third-party images, such as figures from the literature, stock photos, clip art or commercial satellite and map data. We strongly discourage the use or adaptation of previously published images, but if this is unavoidable, please request the necessary rights documentation to re-use such

material from the relevant copyright holders and return this to us when you submit your revised manuscript.

Response #14: No third party images have been used.

Action #14: None needed.

Query #15: DISPLAY ITEMS * Please ensure that figure legend titles are brief - they should not occupy more than one line.

Response #15: Fig 2. Legend was longer than 1 line, thus has been edited, line 557.

Action #15: Edits

Before (Line 557): Fig. 2. Drift tube ion mobility separation (DTIMS) of isomeric HS standards providing unique ATD and CCS values

After: Fig. 2. Drift tube ion mobility separation (DTIMS) of isomeric HS standards

Query #16: * Please define any new abbreviations, symbols or colours present in your figures in the associated legends. Please do not use symbols in your legend, instead please write out the symbols in words (blue circles, red dashed line, etc.).

Response #16: Figure legends checked and modified to conform where necessary.

Action #16: The figure legends have been changed accordingly:

Fig. 1 legend abbreviations; line 546 "ion mobility mass spectrometry (IMMS)", line 548, "collision cross section (CCS)"

Fig. 2 legend abbreviations; line 558 "collision cross section (CCS)"

Fig. 3 legend abbreviations; line 575 "drift tube ion mobility mass spectrometry (DTIMS) and "collision cross sections (CCS)"

Fig. 4 legend abbreviations; line 591 "fibroblast growth factor 1 (FGF1)"

Fig 5. Legend Symbols; Line 601 "#HS1 (black circle with line) and #HS2 (white circle with line)

Query #17: SUPPLEMENTARY INFORMATION * Please replace general citations to the Supplementary Information (e.g. "see Supplementary Information") with specific citations (e.g. "See Supplementary Figure 1", etc.).

Response #17: The text has been edited to conform

Action #17: all Supplementary information has been change to "See Supplementary Table X or See Supplementary Figure Y".

Query #18: SOURCE DATA * Please provide a source data file with the data underlying the reported averages and error bars in Figure 5b.

Response #18: Data attached in the Source Data Excel file. There are 3-9 replicates for Baf3 assay data for Fig4 and Fig 5b. The error bars for Fig 5b were taken from a triple experiment at multiple concentrations.

Action #18: Uploaded Excel file with Baf3 assay data.

Query #19: * Your paper will be accompanied by a two-sentence Editor's summary, of between 250-300 characters including spaces, when it is published on our homepage. Could you please approve the draft summary below or provide us with a suitably edited version.

"Heparan sulfates (HS) contain functionally relevant structural motifs but determining their monosaccharide sequence remains challenging. Here, the authors develop an ion mobility mass spectrometry-based method that allows unambiguous characterization of HS sequences and structure-activity relationships."

Action #19: Approve, the statement the editor suggested is fine.

Query #20: We encourage increased transparency in peer review by publishing the reviewer comments and author rebuttal letters of our research articles, if the authors agree. Such peer review material is made available as a supplementary peer review file. **Please state in the cover letter 'I wish to participate in transparent peer review' if you want to opt in, or 'I do not wish to participate in transparent peer review' if you don't.** Failure to state your preference will result in delays in accepting your paper for publication.

Please note: we allow redactions to authors' rebuttal and reviewer comments in the interest of confidentiality. If you are concerned about the release of confidential data, please let us know specifically what information you would like to have removed. Please note that we cannot incorporate redactions for any other reasons. Reviewer names will be published in the peer review files if the reviewer signed the comments to authors, or if reviewers explicitly agree to release their name.

Response #20: We would like to participate in a transparent peer review.

Action #20: Stated in Letter "I wish to participate in transparent peer review".

Query #21: Springer Nature encourages all authors and reviewers to adopt an Open Researcher and Contributor Identifier (ORCID). ORCID is a community-based initiative that provides an open, non-proprietary and transparent registry of unique identifiers to help disambiguate research contributions. All authors who link their ORCID to their account in our submission system will have their ORCID published on their articles. Please note that this is only possible if ORCIDs are linked prior to acceptance, that is, it is not possible to add ORCIDs at proof.

Please ensure that all co-authors are aware that they can link their ORCIDs, so that it will display on this paper. If they so wish, they must do so before the paper is formally accepted. It will not be possible to add ORCIDs post-acceptance, e.g. at proof. To link an ORCID please follow these instructions:

1. From the home page of the MTS click on 'Modify my Springer Nature account' under 'General tasks'.
2. In the 'Personal profile' tab, click on 'ORCID Create/link an Open Researcher Contributor ID (ORCID)'. This will re-direct you to the ORCID website.
- 3a. If you already have an ORCID account, enter your ORCID email and password and click on 'Authorize' to link your ORCID with your account on the MTS.
- 3b. If you don't yet have an ORCID account, you can easily create one by providing the required information and then clicking on 'Authorize'. This will link your newly created ORCID with your account on the MTS.

Response #21: All authors notified.

Action #21: Authors have been notified of the option to link their ORCID number.

Reviewers' comments:

Reviewer #1 (Remarks to the Author):

This manuscript reports a SIMMS2 approach for HS sequencing, where HS oligosaccharides were dissociated in an ion mobility mass spectrometer, and the CCS values of fragments were measured and searched against the HS fragment CCS library to facilitate structural assignment. The use of fragment CCS values to assist with glycan sequencing was previously reported by Both et al. in their 2013 Nat. Chem. article (<https://www.nature.com/articles/nchem.1817>), and this diminishes the novelty of the current study. Nonetheless, the manuscript under consideration is still the first to report the application of IMMS2 for HS sequencing. A key advantage of SIMMS2 for HS sequencing over the traditional MS2 approach is its ability to define the uronic acid stereochemistry. Although SIMMS2 could POTENTIALLY enable direct and unambiguous sequencing of HS oligosaccharides, its performance was only tested on a few simple HS oligosaccharides. Moreover, the authors failed to adequately address several potential complicating factors that could severely and negatively affect the accuracy and utility of the SIMMS2 approach. A major revision is recommended before the manuscript can be re-evaluated for publication on Nature Communication.

Detailed comments:

(1) 3O-sulfation was not considered, and the authors justified its exclusion on the basis of its rare occurrences. However, 3O-sulfation is often crucial for the proper function of HS, such as the ATIII-binding pentasaccharide that was one of the few defined HS sequence motifs. Omission of 3O-sulfation not only makes the method unsuitable for the analysis of many biologically important HS glycans, but also prevents a more rigorous evaluation of its performance, as increased extent of sulfation can lead to significant challenges in HS sequencing by IMMS2 that are not present in sequencing of HS with a low degree of sulfation. First, sulfo losses become more prominent in highly sulfated HS, making it more difficult to determine the sulfation pattern. Second, a larger number of isomeric building blocks need to be considered, and this increases the chance of isomers sharing similar CCS values. Moreover, it is already fairly easy to determine the sulfation pattern of HS without 3O-sulfation, and sometimes even the uronic acid stereochemistry may be determined in HS with a low degree of sulfation, as demonstrated by the Amster group. Finally, two of the co-authors of this study (Dr. Liu and Dr. Boons) are more than capable of synthesizing HS standards with 3O-sulfation, so the decision to omit 3O-sulfation is really ill-advised.

(2) Many of the HS standards used in this study were modified in the reducing end. Although this allows easy identification of reducing-end fragments for construction of the CCS library, half of the fragments (Y, Z ions) in the library are not useful for analysis of HS oligosaccharides from biological sources, which do not carry these modifications. Worse, biological HS glycans can produce reducing-end and non-reducing end fragments with the same composition. This further complicates CCS matching, and could also negatively impact library expansion.

(3) The presence of internal fragments can also complicate CCS matching, as they may be misinterpreted as B, C, Y, Z ions.

(4) Sulfo loss, which is fairly common with collisional dissociation, can further complicate sulfation pattern determination and CCS matching.

(5) CCS values of some isomers are very close, e.g. CCS values of HS#4 and #5 differ by only 1 Å². This is getting very close to the typical measurement error. Using an IMMS instrument with a higher mobility resolution would be advised.

(6) When the measured CCS value of a fragment can be matched to two isomeric fragments in the library, it may be helpful to consider CCS values of the same fragment in a different charge state. Has the authors consider this possibility?

(7) A more severe issue will arise when two isomeric fragments are produced from a single HS structure: they can be a combination of non-reducing end, reducing-end, internal fragments, or these fragments with sulfo losses. If they have sufficiently close CCS values, the two overlapping peaks will converge into one peak with a centroid value that is different from either fragment, resulting in misidentification by CCS matching.

(8) A single HS glycan or fragment can exist in more than one conformations. This has been

reported in the literature and was also observed in the present study (e.g. HS#18, Figure 2c). Multiple CCS values should be reported in these cases to improve the accuracy of CCS matching.

(9) There is no discussion on whether the source conditions can influence the conformation, and if this can be problematic for CCS matching.

(10) There is a chance that the same fragment from different intact HS structures or from the same HS structure in different charge states have different CCS values (particularly for larger oligosaccharides) because they have different deprotonation sites, and the conformation of the fragments is kinetically trapped.

(11) The discussion on how the uronic acid stereochemistry affects the CCS values seems a bit far-fetched. The trend can be reversed depending on the location of the uronic acid residues, and the nature of the reducing-end modification. The truth is, HS conformation is likely influenced by its charge states and many structural variables, and these effects are not additive. It is probably difficult to predict the CCS values without molecular modeling, and a universal trend is unlikely to exist. As long as the CCS value of a given fragment is constant and can be reliably measured, discussion of trends with these hand-waving arguments is really not necessary.

(12) The data interpretation seems to be manual and subjective. It would be helpful if this can be done automatically, but this is not required for publication.

Reviewer #2 (Remarks to the Author):

The SIMM2 method described by Miller and colleagues represents a timely and significant advance for the field of biologically active heparan sulfate (HS) where the goal has been for many years to link function with structure, which has been brought closer with the detailed description of this novel methodology. In particular, the work described here focuses on the FGF binding and activation activities of oligosaccharides of similar lengths with the difference being a single 2 sulfate group on a central iduronic acid. This is an important and novel finding that differs from other previous studies in the literature showing that 6 sulfate groups are important for activation. The data is shown in supplementary figures but are central to the utility of the SIMM2 method and how it will help to sort out what is a confusing literature. Can these be moved into the main list of figures so that the story flows better and the reader does not have to look for the figures in the supplementary list?

The extensive list of tables is appropriate to be part of the supplementary data as it serves to be an solid reference source for labs wanting to establish the methodology as a part of their experimental workflow. It seems like the addition of the drift tube approach that the variability of the CCS scores between equipment and labs will be minimised. Have the authors looked at the variability between equipment, operators and laboratories. This will be particularly useful for discerning isomeric oligosaccharides without any ambiguity between labs around the world. The data shown in figure 4c does not fit well with the FGF focused story of the paper. Have the authors looked at sequencing the well described ATIII binding pentasaccharide using the SIMM2 methodology? This would be a useful addition to the paper and would support the sequence of the HS1 and HS2 oligosaccharides in that the pentasaccharide sequence would be absent in both. The addition of heparin ATIII binding sequences would also broaden the interest of the readership to include those using or manufacturing heparin for clinical use or developing new heparin-like anticoagulants into the future.

Reviewer #3 (Remarks to the Author):

The article by Miller et al has used negative mode-tandem IM-MS on a drift tube-ion mobility spectrometry (DTIMS) to differentiate multiple oligosaccharides and some cases with sulfate positional isomers. While heparan sulfate (HS) system is important and only few IM-MS-related studies, the technological innovation is rated moderate. Negative mode IM-MS has been reported in reference 21 (Nature 2015) and tandem MS strategy/CID has been used in reference 20 (Nat. Chem. 2014). The basic conception behind the manuscript is also seemingly unclear, as MS/MS-

only experiments can easily distinguish b4/b5 ions as shown in Supplementary Figure 1a due to the differences in mass of b5. As such, it is not entirely clear how IM can add more into MS/MS identification in 2S localization. Although the overall manuscript is very informative, the authors did not articulate sufficiently the technical innovation and advancements of the current study compared to published IM-MS reports on glycan analysis. Some of the major statements were not supported by their data in a convincing way.

Specific Comments:

1. Overall, the method of employing CCS comparison for disaccharide units is appreciated. However, it is suspected CID of various GAGs/HS are likely to result in fragments of odd lengths (e.g. mono- and trisaccharides). For this reason, some discussion as to how you remedy fringe cases where fragment ions do not coincide with disaccharide measurements is of topical interest.
2. The value of direct CCS measurement through DTIMS is clearly felt, but details over the number of replicates used to ensure CCS determinations are reproducible are needed. This is mentioned as a footnote in the supplemental information, but should also be included in the main methods section under "accurate CCS measurements."
3. Consider discussing or describing instances where CCS measurements from biological HS deviated from the standard measurements (or where they were close but not exact) and how any corrections were applied.
4. As the CCS measurement software is home-built, it is important to include algorithm details to enable readers the ability to produce their own such software. Alternatively, the Python code for these measurements should be provided and not dependent on supplemental requests.
5. The idea that "sequence unknown HS structures using data from overlapping standards" is intriguing but also challenging, because even when the CCSs of unknown glycans match well with standards in some way, it does not necessarily mean they have the same structures. This aspect is particularly relevant as glycan subunits can orient in many ways and have many stereoisomers that can behave the same in IM-MS measurements. This caveat needs to be further elaborated with caution.
6. The authors are advised to improve the Introduction part by highlighting the difference in technical aspect of this manuscript compared to previous IM-MS-based glycan sequencing and analysis. This was largely missing from the introduction part.
7. P5: The authors must report uncertainties for all the CCS values as they were used to compare/separate from each of these glycan isomers. In addition, the CCS values in the Results part were not exactly the same with those listed in the corresponding Tables. The authors need to be more rigorous with presenting the data. The conclusion of "IMMS was able to fully distinguish between disaccharides containing the core structures UA-GlcNAc and UA-GlcNS" is inaccurate, as #4 and #5 only showed very small difference in center CCS values with poor separation.
8. P15: The authors mentioned the CCSs values are calculated using a home-developed software. First, the authors need to provide detailed description of the software and the algorithm. As such, readers can follow-up and validate the reports reported in this manuscript. Second, the authors mentioned drift times will need to be extracted manually in some un-perfect cases, which suggests that the authors should report the full CCS distribution rather than the center CCS values which may vary dramatically from one to another experiments and even replicates.
9. Figure 2: Why #9 and #18 have two conformers but #10 and #17 don't? It is surprising that both #10 and #17 showed broader CCS distribution compared to #9 and #18. Can authors provide more explanation/discussion about this result?
10. Figure 4b/c: statistical information is missing.
11. Please double check the reference formats, including refs. 21/47.

Minor/grammatical suggestions:

1. On page four, in the first "Results" paragraph, consider revising "The final library included;" to "The final library included".
2. Add comma to first sentence of last paragraph on page 9: "...analysis into account, the potential...".

List of point-by-point query-response-actions:

Reviewer #1

Query #1: 3O-sulfation was not considered, and the authors justified its exclusion on the basis of its rare occurrences. However, 3O-sulfation is often crucial for the proper function of HS, such as the ATIII-binding pentasaccharide that was one of the few defined HS sequence motifs. Omission of 3O-sulfation not only makes the method unsuitable for the analysis of many biologically important HS glycans, but also prevents a more rigorous evaluation of its performance, as increased extent of sulfation can lead to significant challenges in HS sequencing by IMMS² that are not present in sequencing of HS with a low degree of sulfation. First, sulfo losses become more prominent in highly sulfated HS, making it more difficult to determine the sulfation pattern. Second, a larger number of isomeric building blocks need to be considered, and this increases the chance of isomers sharing similar CCS values. Moreover, it is already fairly easy to determine the sulfation pattern of HS without

3O-sulfation, and sometimes even the uronic acid stereochemistry may be determined in HS with a low degree of sulfation, as demonstrated by the Amster group. Finally, two of the co-authors of this study (Dr. Liu and Dr. Boons) are more than capable of synthesizing HS standards with 3O-sulfation, so the decision to omit 3O-sulfation is really ill-advised.

Response #1: We have now gained access to 3O-sulfated standards and can present additional data.

Action #1: Data with 3O-sulfated stds are now included in Figure 5 (p23), Supplementary Fig. 2 (p5), Supplementary Tables 1 (p9) and Tables 19-25 (p41-47). The following sentences have been added on p6 – line 151, p9 – line 216 and p11 – line 268:

“The 3O-sulfated tetrasaccharides corresponding to the ATIII sites in porcine Δ UA-GlcNS-IdoA2S-GlcNS3S (#30) and Δ UA-GlcNS6S-GlcA-GlcNS3S6S (#31) displayed CCS values of 228 \AA^2 and 229 \AA^2 respectively. Whilst the tetrasaccharide corresponding to the ATIII site in bovine heparin Δ UA-GlcNAc6S-GlcA-GlcNS3S6S (#32) displayed a CCS value of 234 \AA^2 .”

“The classic high affinity antithrombin III binding site responsible for the anticoagulant activity in pharmaceutical heparin is important^{37,38}, therefore CCS values were determined for the enzymatically depolymerised natural structures (#29-#32) and synthesised structures (#33-35) that correlated with the porcine and bovine ATIII binding site (Supplementary Tables 19 to 25). SIMMS² was used to sequence isomeric structures #33 and #34 (Supplementary Table 1) that differed only in the presence of either a 6O-sulfate or a 3O-sulfate on a single glucosamine residue (Supplementary Fig. 2). B fragment ions from #33 and #34 were distinguishable: the B₄ fragment ion of the 6O-sulfated isomer (#33) has a CCS value of 225 \AA^2 , whereas the B₄ fragment of the isomer carrying 3O-sulfation (#34) has a CCS value of 220 \AA^2 (Supplementary Fig. 2 and Supplementary Tables 23 and 24). This data clearly demonstrates the ability of SIMMS² to determine structures containing 3O-sulfate groups.”

“A comparison of tetrasaccharide #30 (from the ATIII binding site) to #HS2 confirmed B₁, B₂, and B₃ fragment ions. The B₄ ion of #30 and #HS2 are isomers, with the former containing a 3O-sulfate and a CCS value of 218 \AA^2 and the latter a 6O-sulfate with a CCS of 225 \AA^2 (Fig. 5 and Supplementary Tables 20 and 68).”

Query #2: Many of the HS standards used in this study were modified in the reducing end. Although this allows easy identification of reducing-end fragments for construction of the CCS library, half of the fragments (Y, Z ions) in the library are not useful for analysis of HS oligosaccharides from biological sources, which do not carry these modifications. Worse, biological HS glycans can produce

reducing-end and non-reducing end fragments with the same composition. This further complicates CCS matching, and could also negatively impact library expansion.

Response #2: The majority of assignments are based on fragments arising from the non-reducing end, which are not affected by the presence of a tag. In addition, odd numbered fragments are informative in distinguishing reducing and non-reducing ends as they differ by mass. Chemically synthesized standards to date carry tags at their reducing end. These structures represent the state-of-the-art for the analysis of structure-activity relationships, therefore it's advantageous and essential to include them in our library. In addition, multiple natural structures with a free reducing end are included in the manuscript. Moreover, the data will be added to the Unicarb KB database, which is open to everyone. If structures with a free non-reducing end or tag become available, the resulting data can be added to the database.

Action #2: None

Query #3: The presence of internal fragments can also complicate CCS matching, as they may be misinterpreted as B, C, Y, Z ions.

Response #3: This is a generic problem with tandem mass spectrometry which might only be solved when isotopically labelled standards become available. Internal fragments are typically formed at higher collision energies, while fragments arising from a single cleavage at the non-reducing side are preferentially formed at low collision energy. Due to the risk of sulfation loss, the collision energy was kept as low as possible. Therefore, the formation of internal fragments is unlikely.

Action #3: None.

Query #4: Sulfo loss, which is fairly common with collisional dissociation, can further complicate sulfation pattern determination and CCS matching.

Response #4: We agree. We found that glycosidic fragment ions showing loss of a sulfate group displayed unique CCS values, and these CCS values were as diagnostic as the glycosidic fragments without sulfation loss for the purposes of sequencing. While the detailed structure of the sulfate-loss fragments remains elusive, they are still highly useful for fingerprinting.

Action #4: A new Supplementary Fig. 1 (p4) demonstrating that fragment ions with loss of a sulfate group are diagnostic for sequencing has been included. Data for fragment ions with sulfation loss for all standards has been included as Supplementary Tables 27-51, (p50-78). Furthermore, a sentence has been added on p8 – line 205 to highlight the value of sulfate loss information.

“In addition, with the aid of CCS values of fragments ions created by loss of a sulfate group from the two tetrasaccharides #12 and #14, we were able to sequence the hexasaccharide structure #16 (Supplementary Fig. 1 and Supplementary Tables 28, 29 and 31).”

Query #5: CCS values of some isomers are very close, e.g. CCS values of HS#4 and #5 differ by only 1 Å². This is getting very close to the typical measurement error. Using an IMMS instrument with a higher mobility resolution would be advised.

Response #5: We agree in principle. Currently, a CCS difference of 0.5-1% is required to confidently identify isomeric structures. The two isomeric standards #4 and #5 may indeed stretch the limits of the method. The resolving power of an instrument and the error associated with the determination of CCS (accuracy) are different. Instruments with higher resolving power do not necessarily provide better accuracy and lower uncertainties. In our method a DTIMS was employed, which does not rely

on calibration. Using the stepped-field method, DTIMS provides absolute CCS values with the highest achievable accuracy.

Action #5: We have removed the interpretation of the CCS difference between standards #4 and #5.

Query #6: When the measured CCS value of a fragment can be matched to two isomeric fragments in the library, it may be helpful to consider CCS values of the same fragment in a different charge state. Has the authors consider this possibility?

Response #6: We agree. The presented Figures show selected charge states with the best separation, however, all charge states are shown in the Supplementary Tables.

Action #6: The following sentence has been modified to clarify this on p5 – line 117:

“The resulting CCS values derived from the 36 HS standards (#1 to #36) and their fragments (Supplementary Tables 1 and 2) demonstrated the ability of IMMS to distinguish between intact structures, different sized fragment ions, charge states and isomeric structures (Supplementary Table 2).”

Query #7: A more severe issue will arise when two isomeric fragments are produced from a single HS structure: they can be a combination of non-reducing end, reducing-end, internal fragments, or these fragments with sulfo losses. If they have sufficiently close CCS values, the two overlapping peaks will converge into one peak with a centroid value that is different from either fragment, resulting in misidentification by CCS matching.

Response #7: We thank the reviewer for this remark. For chemical standards with a tag, the reducing end problem does not arise. For natural structures with a free reducing end this problem – generally present in the tandem MS of oligosaccharides as stated in response #3 – was successfully tackled by comparing multiple CCS values from fragment ions to determine a single structure.

Action #7: The following sentence has been added to the Discussion to clarify this on p12 – line 303:

“Separation of isomeric glycans in some cases is possible, whereas some isomeric glycans display similar CCS values⁴⁷. In the present study, the same phenomenon was observed. Thus, to ensure correct oligosaccharide sequence identification, a comparison of multiple CCS fragment values should be performed, and the interpretation should include information from MS/MS and disaccharide analysis.”

Query #8: A single HS glycan or fragment can exist in more than one conformation. This has been reported in the literature and was also observed in the present study (e.g. HS#19, Figure 2c). Multiple CCS values should be reported in these cases to improve the accuracy of CCS matching.

Response #8: This is an interesting point, since the influence of conformers on the ion mobility separation of biomolecular ions is a field of intense debate. For example, the two peaks in the ATD of Man-9 ions was originally thought to correspond to conformers (Struwe et al). Later it turned out that this interpretation was incorrect and the two species were identified as reducing end anomers (Harvey et al). Oligosaccharide conformers are not thought to interconvert on the time scale of ion mobility separations, nevertheless this is yet to be confirmed. Whenever present, we added CCS values for multiple peaks including #19 in supplementary tables.

Action #8: Corrected information about multiple peaks added.

W. B. Struwe, J. L. Benesch, D. J. Harvey, and K. Pagel. Collisional cross sections of high-mannose N-glycans in commonly observed adduct states – identification of gas-phase conformers unique to $[M - H]^-$ ions. *Analyst*, 2015, 140, 6799.

David J. Harvey and Jodie L. Abrahams. Fragmentation and ion mobility properties of negative ions from N-linked carbohydrates: Part 7. Reduced glycans. *Rapid Commun. Mass Spectrom.* 2016, 30, 627-634

Query #9: There is no discussion on whether the source conditions can influence the conformation, and if this can be problematic for CCS matching.

Response #9: We thank the reviewer for pointing this out. This issue has been addressed in our response #8. In addition, we performed additional experiments to prove that the CCS of ions remains unaltered under very different (soft and harsh) source conditions.

Action #9: We have now included Supplementary Fig. 5 (p8), and the following sentence in the Methods section (p16 – line 416):

“Standard #10 was analysed under soft and harsh source conditions, and demonstrated that the determined CCS value was unchanged (Supplementary Figure 5).”

Query #10: There is a chance that the same fragment from different intact HS structures or from the same HS structure in different charge states have different CCS values (particularly for larger oligosaccharides) because they have different deprotonation sites, and the conformation of the fragments is kinetically trapped.

Response #10: Previous IMS studies on synthetic oligosaccharides have shown that fragment ions exhibit the same CCSs, independently of their origin (Hofmann *Nature* 2015). If deprotomers did interconvert and appeared as separate peaks, they would be added to the database as separate species. Finally, given the rather small size of the ions and their high internal energy following CID, trapping of distinct conformers is rather unlikely and has not been reported to the best of our knowledge for oligosaccharide ions.

Action #10: None.

Hoffmann J, Hahm HS, Seeberger PH, Pagel K. Identification of carbohydrate anomers using ion mobility-mass spectrometry. *Nature*. 2015, 526, 241-244

Query #11. The discussion on how the uronic acid stereochemistry affects the CCS values seems a bit far-fetched. The trend can be reversed depending on the location of the uronic acid residues, and the nature of the reducing-end modification. The truth is, HS conformation is likely influenced by its charge states and many structural variables, and these effects are not additive. It is probably difficult to predict the CCS values without molecular modeling, and a universal trend is unlikely to exist. As long as the CCS value of a given fragment is constant and can be reliably measured, discussion of trends with these hand-waving arguments is really not necessary.

Response #11. We fully agree with the reviewer and understand that our previous discussion was too general and speculative.

Action #11: We have removed these parts from the result and discussion sections.

Query #12: The data interpretation seems to be manual and subjective. It would be helpful if this can be done automatically, but this is not required for publication.

Response #12: All ATDs were analysed using a Gaussian fitting protocol implemented in a Python program written in-house. As such, none of the data interpretation is manual or subjective.

Action #12: We have now included a link to the Python code for CCS value determination in the methods and include the following sentence on p18 – line 465:

“Code availability. The algorithm for processing ion-mobility data (Aprid) was written in-house and used for CCS calculation in an automated fashion. It is implemented using Python 2.7 and can be operated by the user through a graphic interface devised in PyQtDesigner. The main goal of the script is to determine collision cross-sections (CCSs) for specific mass-to-charge (m/z) ratios. For that purpose, the script follows the mathematical approach of solving the Mason-Schamp equation. Details about the code and modules can be found online via github: <https://github.com/waschbaerlauch/CCS-Calculation>.

Reviewer #2.

Query #1: In particular, the work described here focuses on the FGF binding and activation activities of oligosaccharides of similar lengths with the difference being a single 2 sulfate group on a central iduronic acid. This is an important and novel finding that differs from other previous studies in the literature showing that 6 sulfate groups are important for activation. The data is shown in supplementary figures but are central to the utility of the SIMM2 method and how it will help to sort out what is a confusing literature. Can these be moved into the main list of figures so that the story flows better and the reader does not have to look for the figures in the supplementary list?

Response #1: We thank the reviewer for this helpful comment and appreciate the suggestion.

Action #1: The data is now shown as Fig. 4 (p22), and the following paragraph has been added to the Result section to stress this finding (p9 – line 228)

“We used BaF3 cell assays³⁸ to evaluate activation and inhibition of FGFs by HS compounds. First, we validated the assay with heparin and the nonamer compounds #25/26, which interestingly showed that the single 2O-sulfate group introduced in #26 provides weak partial FGF2 activation, while neither structure showed inhibitory activity (Fig. 4b). This is noteworthy since previous results demonstrated that NS and 2OS are required for FGF2 binding, while 6OS is required for optimal promotion of FGF2 cellular signaling^{41, 42}. The nonasaccharide #26 lacks 6OS and still partially promotes FGF2 signalling, demonstrating the utility of complete characterization of sequences by SIMMS2 in deciphering cues for FGF bioactivities.”

And the following paragraph has been added to the Discussion (p12 – line 294):

“We also exploited SIMMS² to sequence an enzymically modified 9mer and showed that the addition of a single 2OS group conferred partial activation of FGF2 (but not FGF1), despite lacking any 6O-sulfation. This illustrates the utility of SIMMS² for defining structure-activity relationships for GAG saccharides and emphasize its potential for the discovery of bioactive motifs with pharmacological potential.”

Query #2: The extensive list of tables is appropriate to be part of the supplementary data as it serves to be a solid reference source for labs wanting to establish the methodology as a part of their experimental workflow. It seems like the addition of the drift tube approach that the variability of the CCS scores between equipment and labs will be minimised. Have the authors looked at the

variability between equipment, operators and laboratories. This will be particularly useful for discerning isomeric oligosaccharides without any ambiguity between labs around the world.

Response #2: This is an excellent point, which has been addressed very systematically by others recently. It was shown that DTIMS can yield CCSs of surprisingly good reproducibility, with inter-laboratory variations below 0.5% (Stow et al).

Ref. Sarah M. Stow, Tim J. Causon, Xueyun Zheng, Ruwan T Kurulugama, Teresa Mairinger, Jody C. May, Emma E Rennie, Erin S. Baker, Richard D. Smith, John A. McLean, Stephan Hann and John C. Fjeldsted. An Interlaboratory Evaluation of Drift Tube Ion Mobility–Mass Spectrometry Collision Cross Section Measurements

Action #2: We have included the following sentence in the Discussion section (p13 – line 336):

“Oligosaccharide CCS values in this study were determined using a single DTIMS instrument. Reproducibility, uncertainty and instrument variability has been addressed previously in an interlaboratory study⁵⁵. For fatty acids and metabolites an average error of $0.27 \pm 0.18\%$ and $0.44 \pm 0.28\%$ was observed. Peptides and proteins showed the largest error with the stepped field method, $0.53 \pm 0.44\%$ and $0.68 \pm 0.36\%$, respectively”.

Query #3: The data shown in figure 4c does not fit well with the FGF focused story of the paper. Have the authors looked at sequencing the well described ATIII binding pentasaccharide using the SIMM2 methodology? This would be a useful addition to the paper and would support the sequence of the HS1 and HS2 oligosaccharides in that the pentasaccharide sequence would be absent in both. The addition of heparin ATIII binding sequences would also broaden the interest of the readership to include those using or manufacturing heparin for clinical use or developing new heparin-like anticoagulants into the future.

Response #3: We agree, and have addressed this under Reviewer 1, Action #1.

Reviewer #3.

Query #1. However, it is suspected CID of various GAGs/HS are likely to result in fragments of odd lengths (e.g. mono- and trisaccharides). For this reason, some discussion as to how you remedy fringe cases where fragment ions do not coincide with disaccharide measurements is of topical interest.

Response #1: We agree and both odd and even numbered fragments are usually observed and all are useful for sequencing.

Action #1: The following sentence has been added to Discussion (p11 – line 288):

“It was identified that odd and even fragments are informative for SIMMS² sequencing”

Query #2. The value of direct CCS measurement through DTIMS is clearly felt, but details over the number of replicates used to ensure CCS determinations are reproducible are needed. This is mentioned as a footnote in the supplemental information, but should also be included in the main methods section under “accurate ccs measurements.”

Response #2: We thank the reviewer for this very helpful comment. The missing information has been added to the manuscript text.

Action #2: The following paragraph has been added to the Methods section p16 – line 420:

“CCS values were determined using the stepped-field method, i.e. by plotting the arrival times (centroid of the best-fit Gaussian) as a function of reciprocal drift voltages. Correlation coefficient (R^2) of the linear fits were typically ≥ 0.9999 . ATDs were recorded at 8 different

voltages ranging from 50 V to 120 V. Each given CCS value is the average of three independent measurements with deviations generally around 0.5%.”

Query #3. Consider discussing or describing instances where CCS measurements from biological HS deviated from the standard measurements (or where they were close but not exact) and how any corrections were applied.

Response #3: Previous IMS studies on synthetic oligosaccharides have shown that fragment ions exhibit the same CCSs, independently of their origin (Hofmann Nature 2015). As such, very little difference is expected for fragments of similar structure arising from different samples. A potential source for differences is the deviation between individual measurements, which on the utilized instrument is typically around 0.5%.

Action #3: None.

Query #4. As the CCS measurement software is home-built, is important to include algorithm details to enable readers the ability to produce their own such software. Alternatively, the Python code for these measurements should be provided and not dependent on supplemental requests.

Response #4: We agree, and the automated Python code script used has been made available.

Action #4: As per Reviewer 1, Action #12.

Query #5: The idea that “sequence unknown HS structures using data from overlapping standards” is intriguing but also challenging, because even when the CCSs of unknown glycans match well with standards in some way, it does not necessarily mean they have the same structures. This aspect is particularly relevant as glycan subunits can orient in many ways and have many stereoisomers that can behave the same in IM-MS measurements. This caveat needs to be further elaborated with caution.

Response #5: The reviewer is correct, the mass and CCS of different structures may indeed coincide. However, in our method of structural assignment is based on the analysis of multiple fragments from one precursor, circumventing this problem.

Action #5: This point is addressed by the sentence added to Discussion on p12 – line 303 (Reviewer 1, Action #7).

Query #6. The authors are advised to improve the Introduction part by highlighting the difference in technical aspect of this manuscript compared to previous IM-MS-based glycan sequencing and analysis. This was largely missing from the introduction part.

Response #6: We thank the reviewer for pointing this out.

Action #6: The paragraph has been added to the introduction (p3 – line 72).

“IMMS of glycans has demonstrated the ability to characterise isomeric arrangements of glycan building blocks (mannose, galactose, etc), but also connectivity (glycosidic bond linkage, sialic acids) and configuration (α and β anomers)¹⁵⁻¹⁸. GAGs possess a linear sequence with no branching, and connectivity is singular to the GAG family. However, GAGs pose different challenges for IMMS analysis. Their complexity is a result of heterogeneity from sulfation and epimerisation, and extended linear units.”

15. Gray JC, Lukasz G, Migas, Barran P E., 1 Pagel K, Seeberger PH., Evers CE., Boons Geert-J, Pohl NLB., Compagnon I, Widmalm G, and Flitsch S. *ESI: Advancing Solutions to the Carbohydrate Sequencing Challenge*. **Journal of the American Chemical Society**, 2019, 141, 37, 14463-14479.
16. Gray, Christopher; Schindler, Baptiste; Migas, Lukasz; Picmanova, Martina; Allouche, Abdul-Rahman; Green, Anthony; Mandal, Santanu; Motawia, Mohammed; Sánchez-Pérez, Raquel; Bjarnholt, Nanna; Møller, Birger; Rijs, Anouk; Barran, Perdita; Compagnon, Isabelle; Evers, Claire; Flitsch, Sabine. "Bottom-up elucidation of glycosidic bond stereochemistry". *Analytical Chemistry*, 2017, 89 (8), 4540-4549.
17. H. Hinneburg, J. Hofmann, W. B. Struwe, A. Thader, F. Altmann, D. Varón Silva, P. H. Seeberger, K. Pagel and D. Kolarich. Distinguishing *N*-acetylneuraminic acid linkage isomers on glycopeptide by ion mobility-mass spectrometry. *Chemical Communications*. 2016, 52, 4381-43-84.
18. Both, P.; Green, A. P.; Gray, C. J.; Sardzik, R.; Voglmeir, J.; Fontana, C.; Austeri, M.; Rejzek, M.; Richardson, D.; Field, R. A.; Widmalm, G.; Flitsch, S. L.; Evers, C. E. "Discrimination of epimeric glycans and glycopeptides using IM-MS and its potential for carbohydrate sequencing" *NATURE CHEMISTRY*. 6(1); 65 – 74.

Query #7. P5: The authors must report uncertainties for all the CCS values as they were used to compare/separate from each of these glycan isomers. In addition, the CCS values in the Results part were not exactly the same with those listed in the corresponding Tables. The authors need to be more rigorous with presenting the data. The conclusion of "IMMS was able to fully distinguish between disaccharides containing the core structures UA-GlcNAc and UA-GlcNS" is inaccurate, as #4 and #5 only showed very small difference in center CCS values with poor separation.

Response #7: We thank the reviewer for this helpful remark. We have removed the interpretation of the CCS difference between standards #4 and #5. Moreover, CCS values were carefully checked and corrected when needed and errors associated with the measurements are now included.

Action #7: Corrected and errors now included in revised Tables.

Query #8. P15: The authors mentioned the CCSs values are calculated using a home-developed software. First, the authors need to provide detailed description of the software and the algorithm. As such, readers can follow-up and validate the reports reported in this manuscript. Second, the authors mentioned drift times will need to be extracted manually in some un-perfect cases, which suggests that the authors should report the full CCS distribution rather than the center CCS values which may vary dramatically from one to another experiments and even replicates.

Response #8: We thank the reviewer for these comments. The software is publicly available for validation purposes. It would indeed be desirable to have CCS distributions of all species, but this turned out to be highly impractical especially for the construction of larger databases.

Action #8: Details of the software access were addressed under Reviewer 1, Action #12.

Query #9. Figure 2: Why #9 and #19 have two conformers but #10 and #20 don't? It is surprising that both #10 and #20 showed broader CCS distribution compared to #9 and #19. Can authors provide more explanation/discussion about this result?

Response #9: We thank the reviewer for this helpful comment, which is in line with comments raised by Reviewer 1, query #8. Some of the fragments indeed show multimodal ATDs. However, based on previous data we cannot unambiguously assign them as conformers. In the context of GAGs, deprotonation isomers are instead more likely (see *IR action spectroscopy of glycosaminoglycan oligosaccharides*; DOI: 10.1007/s00216-019-02327-7).

Action #9: None.

Query #10. Figure 5b/c: statistical information is missing.

Response #10: Thank you and corrected

Action #10: Statistical data has been added to Fig.4 (p22 – line 528), Fig.5 (p23 – line 534) and methods (p18 – line 457).

Query #11. Please double check the reference formats, including refs. 24/48.

Response #11: Thank you and corrected

Action #11: Page numbers have been added to references 24 and 48

Query #12: On page four, in the first “Results” paragraph, consider revising “The final library included;” to “The final library included”.

Response #12: Thank you. (p4 - line 99)

Action #12: Corrected to “The final library included”

Query #13: Add comma to first sentence of last paragraph on page 9: “...analysis into account, the potential...”.

Response #13: Thank you. (p10 - line 248)

Action #13: Corrected to “...analysis into account, the potential...”